# Information Theoretic Regret Bounds
# for Online Nonlinear Control

**Sham Kakade**[1,2]                    **Akshay Krishnamurthy**[2]

**Kendall Lowrey**[1]          **Motoya Ohnishi**[1]          **Wen Sun**[3]
[1]University of Washington    [2]Microsoft Research NYC    [3]Cornell University

## Abstract

This work studies the problem of sequential control in an unknown, *nonlinear* dynamical system, where we model the underlying system dynamics as an unknown function in a known Reproducing Kernel Hilbert Space. This framework yields a general setting that permits discrete and continuous control inputs as well as non-smooth, non-differentiable dynamics. Our main result, the Lower Confidence-based Continuous Control (LC[3]) algorithm, enjoys a near-optimal $O(\sqrt{T})$ regret bound against the optimal controller in episodic settings, where $T$ is the number of episodes. The bound has *no* explicit dependence on dimension of the system dynamics, which could be infinite, but instead only depends on information theoretic quantities. We empirically show its application to a number of *nonlinear* control tasks and demonstrate the benefit of exploration for learning model dynamics.

## 1   Introduction

The control of uncertain dynamical systems is one of the central challenges in Reinforcement Learning (RL) and continuous control, and recent years has seen a number of successes in demanding sequential decision making tasks ranging from robotic hand manipulation [Todorov et al., 2012, Al Borno et al., 2012, Kumar et al., 2016, Tobin et al., 2017, Lowrey et al., 2018, Akkaya et al., 2019] to game playing [Silver et al., 2016, Bellemare et al., 2016, Pathak et al., 2017, Burda et al., 2018]. The predominant approaches here are either based on reinforcement learning or continuous control (or a mix of techniques from both domains).

With regards to provably correct methods which handle both the learning and approximation in unknown, complex environments, and achieve optimality guarantees, the body of results in the reinforcement learning literature [Russo and Van Roy, 2013, Jiang et al., 2017, Sun et al., 2019, Agarwal et al., 2019a] is more mature than in the continuous controls literature. In fact, only relatively recently has there been provably correct methods (and sharp bounds) for the learning and control of the Linear Quadratic Regulator (LQR) model [Mania et al., 2019, Simchowitz and Foster, 2020, Abbasi-Yadkori and Szepesvári, 2011], arguably the most basic model due to having globally *linear* dynamics.

While Markov Decision Processes provide a very general framework after incorporating continuous states and actions into the model, there are a variety of reasons to directly consider learning in continuous control settings: even the simple LQR model provides a powerful framework when used for *locally* linear planning [Ahn et al., 2007, Todorov and Li, 2005, Tedrake, 2009, Perez et al., 2012]. More generally, continuous control problems often have continuity properties with respect

---

Project page: https://sites.google.com/view/lc3algorithm/

to the underlying "disturbance" (often modeled as statistical additive noise), which can be exploited for fast path planning algorithms [Jacobson and Mayne, 1970, Williams et al., 2017]; analogous continuity properties are often not leveraged in designing provably correct RL models (though there are a few exceptions, e.g. [Kakade et al., 2003]). While LQRs are a natural model for continuous control, they are prohibitive for a variety of reasons: LQRs rarely provide good *global* models of the system dynamics, and, furthermore, naive random search suffices for sample efficient learning of LQRs [Mania et al., 2019, Simchowitz and Foster, 2020] — a strategy which is unlikely to be effective for the learning and control of more complex nonlinear dynamical systems where one would expect strategic exploration to be required for sample efficient learning (just as in RL, e.g. see Kearns and Singh [2002], Kakade [2003]).

This is the motivation for this line of work, where we focus directly on the sample efficient learning and control of an *unknown*, nonlinear dynamical system, under the assumption that the mean dynamics live within some known Reproducing Kernel Hilbert Space.

**The Online Nonlinear Control Problem.** This work studies the following nonlinear control problem, where the nonlinear system dynamics are described, for $h \in \{0, 1, \ldots H - 1\}$, by

$$x_{h+1} = f(x_h, u_h) + \epsilon, \text{ where } \epsilon \sim \mathcal{N}(0, \sigma^2 I)$$

where the state $x_h \in \mathbb{R}^{d_{\mathcal{X}}}$; the control $u_h \in \mathcal{U}$ where $\mathcal{U}$ may be an arbitrary set (not necessarily even a vector space); $f : \mathcal{X} \times \mathcal{U} \to \mathcal{X}$ is assumed to live within some known Reproducing Kernel Hilbert Space; the additive noise is assumed to be independent across timesteps.

Specifically, the model considered in this work was recently introduced in Mania et al. [2020], which we refer to as the *kernelized nonlinear regulator* (KNR) for the infinite dimensional extension. The KNR model assumes that $f$ lives in the RKHS of a known kernel $K$. Equivalently, the primal version of this assumption is that:

$$f(x, u) = W^\star \phi(x, u)$$

for some known function $\phi : \mathcal{X} \times \mathcal{U} \to \mathcal{H}$ where $\mathcal{H}$ is a Hilbert space (either finite or countably infinite dimensional) and where $W^\star$ is a linear mapping. Given an immediate cost function $c : \mathcal{X} \times \mathcal{U} \to \mathbb{R}^+$ (where $\mathbb{R}^+$ is the non-negative real numbers), the KNR problem can be described by the following optimization problem:

$$\min_{\pi \in \Pi} J^\pi(x_0; c) \text{ where } J^\pi(x_0; c) = \mathbb{E}\left[\sum_{h=0}^{H-1} c(x_h, u_h)\middle| \pi, x_0\right]$$

where $x_0$ is a given starting state; $\Pi$ is some set of feasible controllers; and where a controller (or a policy) is a mapping $\pi : \mathcal{X} \times \{0, \ldots H - 1\} \to \mathcal{U}$. We denote the best-in-class cumulative cost as $J^\star(x_0; c) = \min_{\pi \in \Pi} J^\pi(x_0; c)$. Given any model parameterization $W$, we denote $J^\pi(x_0; c, W)$ as the expected total cost of $\pi$ under the dynamics $W\phi(x, u) + \epsilon$.

It is worthwhile to note that this KNR model is rather general in the following sense: the space of control inputs $\mathcal{U}$ may be either discrete or continuous; and the dynamics $f$ need not be a smooth or differentiable function in any of its inputs. A more general version of this problem, which we leave for future work, would be where the systems dynamics are of the form $x_{h+1} = f_h(x_h, u_h, \epsilon_h)$, in contrast to our setting where the disturbance is due to additive Gaussian noise.

We consider an online version of this KNR problem: the objective is to sequentially optimize a sequence of cost functions where the nonlinear dynamics $f$ are not known in advance. We assume that the learner knows the underlying Reproducing Kernel Hilbert Space. In each episode $t$, we observe an instantaneous cost function $c^t$; we choose a policy $\pi^t$; we execute $\pi^t$ and observe a sampled trajectory $x_0, u_0, \ldots, x_{H-1}, u_{H-1}$; we incur the cumulative cost under $c^t$. Our goal is to minimize the sum of our costs over $T$ episodes. In particular, we desire to execute a policy that is nearly optimal for every episode.

A natural performance metric in this context is our cumulative regret, the increase in cost due to not knowing the nonlinear dynamics beforehand, defined as:

$$\text{REGRET}_T = \sum_{t=0}^{T-1} \sum_{h=0}^{H-1} c^t(x_h^t, u_h^t) - \sum_{t=0}^{T-1} \min_{\pi \in \Pi} J^\pi(x_0; c^t)$$

where $\{x_h^t\}$ is the observed states and $\{u_h^t\}$ is the observed sequence of controls. A desirable asymptotic property of an algorithm is to be no-regret, i.e. the time averaged version of the regret goes to $0$ as $T$ tends to infinity.

**Our Contributions.** The first set of provable results in this setting, for the finite dimensional case and for the problem of system identification, was provided by Mania et al. [2020]. Our work focuses on regret, and we provide the Lower Confidence-based Continuous Control (LC$^3$) algorithm, which enjoys a $O(\sqrt{T})$ regret bound. We provide an informal version of our main result, specialized to the case where the dimension of the RKHS is finite and the costs are bounded.

**Theorem 1.1** (Informal statement; finite dimensional case with bounded features). *Consider the special case where: $c^t(x, u) \in [0, 1]$; $d_\phi$ is finite (with $d_\mathcal{X} + d_\phi \geq H$); and $\phi$ is uniformly bounded, with $\|\phi(x, u)\|_2 \leq B$; The LC$^3$ algorithm enjoys the following expected regret bound:*

$$\mathbb{E}_{\mathrm{LC}^3}\left[\mathrm{REGRET}_T\right] \leq \widetilde{O}\left(\sqrt{d_\phi(d_\mathcal{X} + d_\phi)H^3 T} \cdot \log\left(1 + \frac{B^2\|W^\star\|_2^2}{\sigma^2}\right)\right),$$

*where $\widetilde{O}(\cdot)$ notation drops logarithmic factors in $T$ and $H$.*

There are a number of notable further contributions with regards to our work:

- (*Dimension and Horizon Dependencies*) Our general regret bound has *no* explicit dependence on dimension of the system dynamics (the RKHS dimension), which could be infinite, but instead only depends on information theoretic quantities; our horizon dependence is $H^3$, which we conjecture is near optimal. It is also worthwhile noting that our regret bound is only logarithmic in $\|W^\star\|_2$ and $\sigma^2$.

- (*Localized rates*) In online learning, it is desirable to obtain improved rates if the loss of the "best expert" is small, e.g. in our case, if $J^\star(x_0; c^t)$ is small. Under a bounded coefficient of variation condition (which holds for LQRs and may hold more generally), we provide an improved regret bound whose leading term regret depends linearly on $J^\star$.

- (*Moment bounds and LQRs*) Our regret bound does not require bounded costs, but instead only depends on second moment bounds of the realized cumulative cost, thus making them applicable to LQRs, as a special case.

- (*Empirical evaluation:*) Coupled with the right features (e.g., kernels), our method provides competitive results in common continuous control benchmarks, exploration tasks, and complex control problems such as dexterous manipulation.

While our techniques utilize methods developed for the analysis of linear bandits [Dani et al., 2008, Abbasi-Yadkori et al., 2011] and Gaussian process bandits [Srinivas et al., 2009], there are a number of new technical challenges to be addressed with regards to the multi-step extension to Reinforcement Learning. In particular, some nuances for the more interested reader: we develop a stopping time martingale to handle the unbounded nature of the (realized) cumulative costs; we develop a novel way to handle Gaussian smoothing through the chi-squared distance function between two distributions; our main technical lemma is a "self-bounding" regret bound that relates the instantaneous regret on any given episode to the second moment of the stochastic process.

**Notation.** We let $\|x\|_2$, $\|M\|_2$, and $\|M\|_F$ refer to the Euclidean norm, the spectral norm, and the Frobenius norm, respectively, of a vector $x$ and a matrix $M$.

## 2 Related Work

The first set of provable results with regards to this nonlinear control model was provided by Mania et al. [2020], who studied the problem of system identification in a finite dimensional setting (we discuss these results later). While not appearing with this name, a Gaussian process version of this model was originally considered by Deisenroth and Rasmussen [2011], without sample-efficiency guarantees. More generally, most model-based RL/controls algorithms do not explicitly address the exploration challenge nor do they have guarantees on the performance of the learned policy [Deisenroth and Rasmussen, 2011, Levine and Abbeel, 2014, Chua et al., 2018, Kurutach et al.,

---

**Algorithm 1** Lower Confidence-based Continuous Control (LC$^3$)

---

**Require:** Policy class $\Pi$; regularizer $\lambda$; confidence parameter $C_1$ (see Equation 3.3).
 1: Initialize BALL$^0$ to be any set containing $W^\star$.
 2: **for** $t = 0 \dots T$ **do**
 3:      $\pi^t = \arg\min_{\pi \in \Pi} \min_{W \in \text{BALL}^t} J^\pi(x_0; c^t, W)$
 4:      Execute $\pi^t$ to sample a trajectory $\tau^t := \{x_h^t, u_h^t, c_h^t, x_{h+1}^t\}_{h=0}^{H-1}$
 5:      Update BALL$^{t+1}$ (as specified in Equation 3.2).
 6: **end for**

---

2018, Nagabandi et al., 2018, Luo et al., 2018, Ross and Bagnell, 2012]. Departing from these works, we focus on provable sample efficient regret bounds and strategic exploration in model-based learning in the kernelized nonlinear regulator.

Among provably efficient model-based algorithms, works from Sun et al. [2019], Osband and Van Roy [2014], Ayoub et al. [2020], Lu and Van Roy [2019] are the most related to our work. While these works are applicable to certain linear structures, their techniques do not lead to the results herein: even for the special case of LQRs, they do not address the unbounded nature of the costs, where there is more specialized analysis [Mania et al., 2019, Cohen et al., 2019, Simchowitz and Foster, 2020]; these results do not provide techniques for sharp leading order dependencies like in our regret results (and the former three do not handle the infinite dimensional case); they also do not provide techniques for localized regret bounds, like those herein which depend on $J^\star$. A few more specific differences: the model complexity measure Witness Rank from Sun et al. [2019] does contain the kernelized nonlinear regulator if the costs were bounded and the dimensions were finite; Osband and Van Roy [2014] considers a setting where the model class has small Eluder dimension, which does not apply to the infinite-dimensional settings that we consider here; Lu and Van Roy [2019] presents a general information theoretic framework providing results for tabular and factor MDPs. Chowdhury and Gopalan [2019] considers kernelized MDPs and directly assumes a Lipschitz condition on the one step future value function. Concurrently, Curi et al. [2020] considers Gaussian Process Model-based RL but explicitly assumes Lipschitz continuity in the learned models and reward functions and the regret scales exponentially with respect to Lipschitz constants. There are numerous technical challenges addressed in this work which may be helpful for further analysis of models in continuous control problems.

Another family of related work provides regret analyses of online LQR problems. There are a host of settings considered: unknown stochastic dynamics [Abbasi-Yadkori and Szepesvári, 2011, Dean et al., 2018, Mania et al., 2019, Cohen et al., 2019, Simchowitz and Foster, 2020]; adversarial noise (or adversarial noise with unknown mean dynamics) [Agarwal et al., 2019b, Hazan et al., 2019]; changing cost functions with known dynamics [Cohen et al., 2018, Agarwal et al., 2019c]. For the case of unknown (stochastic) dynamics, our online KNR problem is more general than these works, due to a more general underlying model; one distinction is that many of these prior works on LQRs consider the regret on a single trajectory, under stronger stability and mixing assumptions of the process. This is an interesting direction for future work (see Section 5).

On the system identification side, Mania et al. [2020] provides the first sample complexity analysis for finite dimensional KNRs under assumptions of reachability and bounded features. The work of Mania et al. [2020] is an important departure from the aforementioned model-based theoretical results [Sun et al., 2019, Osband and Van Roy, 2014, Ayoub et al., 2020, Lu and Van Roy, 2019]; the potentially explosive nature of the system dynamics makes system ID challenging, and Mania et al. [2020] directly addresses this through various continuity assumptions on the dynamics. One notable aspect of our work is that it permits both an unbounded state and unbounded features. The KNR also has been used in practice for system identification [Ng et al., 2006, Abbeel and Ng, 2005].

## 3 Main Results

### 3.1 The Lower Confidence-based Continuous Control algorithm

LC$^3$ is based on "optimism in the face of uncertainty," which is described in Algorithm 1. At episode $t$, we use all previous experience to define an uncertainty region (an ellipse). The center of

this region, $\overline{W}^t$, is the solution of the following regularized least squares problem:

$$\overline{W}^t = \arg\min_W \sum_{\tau=0}^{t-1} \sum_{h=0}^{H-1} \|W\phi(x_h^\tau, u_h^\tau) - x_{h+1}^\tau\|_2^2 + \lambda\|W\|_F^2, \tag{3.1}$$

where $\lambda$ is a parameter, and the shape of the region is defined through the feature covariance:

$$\Sigma^t = \lambda I + \sum_{\tau=0}^{t-1} \sum_{h=0}^{H-1} \phi(x_h^\tau, u_h^\tau)(\phi(x_h^\tau, u_h^\tau))^\top, \text{ with } \Sigma^0 = \lambda I.$$

The uncertainty region, or confidence ball, is defined as:

$$\text{Ball}^t = \left\{ W \,\Big|\, \left\| \left(W - \overline{W}^t\right)\left(\Sigma^t\right)^{1/2} \right\|_2^2 \leq \beta^t \right\}, \tag{3.2}$$

where recall that $\|M\|_2$ denotes the spectral norm of a matrix $M$ and where

$$\beta^t := C_1\left( \lambda\sigma^2 + \sigma^2\left(d_{\mathcal{X}} + \log\left(t\det(\Sigma^t)/\det(\Sigma^0)\right)\right)\right), \tag{3.3}$$

with $C_1$ being a parameter of the algorithm.

At episode $t$, the LC$^3$ algorithm will choose an optimistic policy in Line 3 of Algorithm 1. Solving this optimistic planning problem in general is NP-hard [Dani et al., 2008]. Given this computational hardness, we focus on the statistical complexity and explicitly assume access to the following computational oracle:

**Assumption 1** (Black-box computation oracle). *We assume access to an oracle that implements Line 3 of Algorithm 1.*

We leave to future work the question of finding reasonable approximation algorithms, though we observe that a number of effective heuristics may be available through gradient based methods such as DDP [Jacobson and Mayne, 1970], iLQG [Todorov and Li, 2005] and CIO [Mordatch et al., 2012], or sampling based methods, such as MPPI [Williams et al., 2017] and DMD-MPC [Wagener et al., 2019]. In particular, these planning algorithms are natural to use in conjunction with Thompson Sampling [Thompson, 1933, Osband and Van Roy, 2014], i.e. we sample $W^t$ from $\mathcal{N}(\overline{W}^t, (\Sigma^t)^{-1})$ and then compute and execute the corresponding optimal policy $\pi^t = \arg\min_{\pi\in\Pi} J^\pi(x_0; c^t, W^t)$ using a planning oracle. While we focus on the frequentist regret bounds, we conjecture that a Bayesian regret bound for the Thompson sampling algorithm, should be achievable using the techniques we develop herein, along with now standard techniques for analyzing the Bayesian regret of Thompson sampling (e.g. see Russo and Van Roy [2014]).

### 3.2 Information Theoretic Regret Bounds

In this section, we analyze the regret of Algorithm 1. Following Srinivas et al. [2009], let us define the (expected) Maximum Information Gain as:

$$
\begin{aligned}
\gamma_T(\lambda) &:= \max_{\mathcal{A}} \mathbb{E}_{\mathcal{A}}\left[ \log\left( \det\left(\Sigma^T\right)/\det\left(\Sigma^0\right)\right)\right] \\
&= \max_{\mathcal{A}} \mathbb{E}_{\mathcal{A}}\left[ \log\det\left( I + \frac{1}{\lambda}\sum_{t=0}^{T-1}\sum_{h=0}^{H-1} \phi(x_h^t, u_h^\tau)(\phi(x_h^t, u_h^\tau))^\top\right)\right],
\end{aligned}
$$

where the max is over algorithms $\mathcal{A}$, where an algorithm is a mapping from the history before episode $t$ to the next policy $\pi_t \in \Pi$.

**Remark 3.1.** (Finite dimensional RKHS and RBF Kernel) For $\phi \in \mathbb{R}^{d_\phi}$, with $\|\phi(x, u)\| \leq B \in \mathbb{R}^+$ for all $(x, u)$, then $\gamma_T(\lambda)$ will be $O(d_\phi \log(1 + THB^2/\lambda)$ (see Lemma C.5). Another example is when $\phi$ is infinite dimensional and corresponds to RBF kernel. In this case, assume $u \in \mathbb{R}^{d_u}$, we have $\gamma_T(\lambda) = O(\sqrt{T\ln(T)^{d_x+d_u+1}})$. Furthermore, it may be the case that $\gamma_T(\lambda) \ll d_\phi$ if the eigenspectrum of the covariance matrices of the policies tend to concentrate in a lower dimensional subspace. See Srinivas et al. [2009] for details and for how $\gamma_T(\lambda)$ scales for a number of popular kernels.

**The General Case, with Bounded Moments**

**Assumption 2.** *(Bounded second moments at $x_0$) Assume that $c^t$ is a non-negative function for all $t$ and that the realized cumulative cost, when starting from $x_0$, has uniformly bounded second moments, over all policies and cost functions $c^t$. Precisely, suppose for every $c^t$,*

$$\sup_{\pi \in \Pi} \mathbb{E}\left[ \left( \sum_{h=0}^{H-1} c^t(x_h, u_h) \right)^2 \,\middle|\, x_0, \pi \right] \leq V_{\max}.$$

This assumption is substantially weaker than the standard bounded cost assumption used in prior model-based RL works (e.g., [Sun et al., 2019, Lu and Van Roy, 2019]), and Lipschitz assumption in terms of value functions (e.g., [Osband and Van Roy, 2014, Chowdhury and Gopalan, 2019]), furthermore, the assumption only depends on the starting $x_0$ as opposed to a uniform bound over the state space. For special case of LQR, this corresponds to restricting $\Pi$ to be the class containing all stable linear controllers.

**Theorem 3.2** (LC$^3$ regret bound). *Suppose Assumptions 1 and 2 hold. Set $\lambda = \frac{\sigma^2}{\|W^\star\|_2^2}$ and define*

$$\widetilde{d}_T^2 := \gamma_T(\lambda) \cdot \big( \gamma_T(\lambda) + d_{\mathcal{X}} + \log(T) + H \big).$$

*There exist constants $C_1, C_2 \leq 20$ such that if LC$^3$ (Alg. 1) is run with input parameters $\lambda$ and $C_1$ (in Equation 3.3), then following regret bound holds for all $T$,*

$$\mathbb{E}_{\mathrm{LC}^3}\left[\mathrm{REGRET}_T\right] \leq C_2 \, \widetilde{d}_T \sqrt{V_{\max} H T}.$$

While the above regret bound is applicable to the infinite dimensional RKHS setting and does not require uniformly bounded features $\phi$, it is informative to specialize the regret bound to the finite dimensional case with bounded features.

**Corollary 3.3** (LC$^3$ Regret for finite dimensional, bounded features). *Suppose that Assumptions 1 and 2 hold; $d_\phi$ is finite; and that $\phi$ is uniformly bounded, with $\|\phi(x,u)\|_2 \leq B$. Under the same parameter choices as in Theorem 3.2, we have, for all $T$,*

$$\mathbb{E}_{\mathrm{LC}^3}\left[\mathrm{REGRET}_T\right] \leq C_2 \sqrt{d_\phi \Big(d_\phi + d_{\mathcal{X}} + \log(T) + H \Big) V_{\max} H T} \cdot \log\left(1 + \frac{B^2 \|W^\star\|_2^2}{\sigma^2} \frac{TH}{d}\right).$$

The above immediately follows from a bound on the finite dimensional information gain (see Lemma C.5).

A few remarks are in order:

**Remark 3.4** (Logarithmic parameter dependencies). It is worthwhile noting that our regret bound has only logarithmic dependencies $\|W^\star\|_2$ and $\sigma^2$. Furthermore, in the case of finite dimensional and bounded features, the bound is also only logarithmic in the bound $B$.

**Remark 3.5** (Dimension and horizon dependencies). For the special case with bounded $c^t(x, u) \in [0, 1]$, bounded $\phi \in \mathbb{R}^{d_\phi}$, and $d_\phi + d_{\mathcal{X}} \geq H$, LC$^3$ has a regret bound of $\widetilde{O}(\sqrt{d_\phi(d_\phi + d_{\mathcal{X}})H^3 T})$. Our dimension dependence matches the lower bounds in [Dani et al., 2008] for linear bandits (where $H = 1$ and $d_{\mathcal{X}} = 1$). Furthermore, for fixed dimension, one might expect an $\Omega(\sqrt{H^2 T})$ lower bound based on results for tabular MDPs (see Azar et al. [2017], Dann and Brunskill [2015]). Obtaining sharp lower bounds is an important direction for future work.

**Remark 3.6** (Linear Quadratic Regulators (LQR) as a special case). Our model generalizes the Linear Quadratic Regulator (LQR). Specifically, we can set $\phi(x, u) = [x^\top, u^\top]^\top$, $c(x, u) = x^\top Q x + u^\top R u$ with $Q$ and $R$ being some PSD matrix. We can consider a policy class to be a (subset) of all linear controls, i.e., $\Pi = \{\pi : u = Kx, K \in \mathcal{K} \subset \mathbb{R}^{d_u \times d_{\mathcal{X}}}\}$.

Consider the case where $d_{\mathcal{X}} = d_u = d$ (with $d > H$) and the policy class consists of controllable policies (e.g. see Cohen et al. [2019]). Here, our regret scales as $\widetilde{O}\left(\sqrt{H^3 d^4 T}\right)$ ( since $V_{\max} = O(Hd^2)$, e.g. see Simchowitz and Foster [2020]). While our rate is a factor of $\sqrt{d}$ larger than the minimax regret bound for an LQR [Simchowitz and Foster, 2020], which is $\Omega(\sqrt{d^3 T})$, our results also apply to non-linear settings, as opposed to the globally linear LQR setting.

**The Stabilizing Case, with Bounded Coefficient of Variation**

In many cases of practical interest, the optimal cost $J^\star(x_0; c)$ may be substantially less than the cost of other controllers, i.e. $J^\star(x_0; c) \ll \max_{\pi \in \Pi} J^\pi(x_0; c) < \sqrt{V_{\max}}$. In such cases, one might hope for an improved regret bound for sufficiently large $T$. We show that this is the case provided our policy class satisfies a certain bounded coefficient of variation condition, which holds for LQRs. Recall the *coefficient of variation* of a random variable is defined as the ratio of the standard deviation to mean.

**Assumption 3** (Bounded coefficient of variation at $x_0$). *Assume that the realized cumulative cost, when starting from $x_0$, has a uniformly bounded coefficient of variation. Specifically, assume there exists an $\alpha \in \mathbb{R}^+$, such that for every $c^t$ and all $\pi \in \Pi$,*

$$\mathbb{E}\left[ \left( \sum_{h=0}^{H-1} c^t(x_h, u_h) \right)^2 \,\Bigg|\, x_0, \pi \right] \leq \left( \alpha \, J^\pi(x_0; c^t) \right)^2.$$

**Remark 3.7** ($\alpha$ for LQRs). It is straightforward to verify that Assumption 3 is satisfied in LQRs (with linear controllers) with $\alpha^2 = 3$.

Under this assumption, we can get a regret bound with a leading order term dependent on $J^\star$. The lower order term will depend on a higher moment version of the information gain, defined as follows:

$$\gamma_{2,T}(\lambda) \quad := \quad \max_{\mathcal{A}} \mathbb{E}_{\mathcal{A}} \left[ \left( \log \left( \det \left( \Sigma^{T-1} \right) / \det \left( \Sigma^0 \right) \right) \right)^2 \right].$$

Again, for a finite dimensional RKHS with features whose norm bounded is by $B$, then $\gamma_{2,T}$ will also be $O((d_\phi \log(1 + THB^2/\lambda))^2)$ (see Remark 3.1 and Lemma C.5).

**Theorem 3.8** ($J^\star$ regret bound). *Suppose that Assumptions 1, 2, and 3 hold and that for all $t$, $J^\star(x_0; c^t) \leq J^\star$. Again, set $\lambda = \frac{\sigma^2}{\|W^\star\|_2^2}$ and define $\widetilde{d}_T$ as in Theorem 3.2. There exist absolute constants $C_1, C_2$ such that if LC³ (Alg. 1) is run with input parameters $C_1$ and $\lambda$, then the following regret bound holds for all $T$,*

$$\mathbb{E}_{\mathrm{LC}^3} \left[ \mathrm{REGRET}_T \right] \leq C_2 \left( \alpha J^\star \widetilde{d}_T \sqrt{HT} + \alpha H \sqrt{V_{\max}} \left( \widetilde{d}_T^2 + \gamma_{2,T}(\lambda) \right) \right).$$

See the Discussion (Section 5) for comments on improving the $J^\star$ dependence to $\sqrt{J^\star}$.

### 3.3 Proof Techniques

A key technical, "self-bounding" lemma in our proof bounds the difference in cost under two different models, i.e. $J^\pi(x; c, W^\star) - J^\pi(x; c, W)$, in terms of the second moment of the cumulative cost itself, i.e. in terms of $V^\pi(x; c, W^\star)$, where

$$V^\pi(x; c, W^\star) := \mathbb{E}\left[ \left( \sum_{h=0}^{H-1} c(x_h, u_h) \right)^2 \,\Bigg|\, x_0 = x, \pi, W^\star \right].$$

**Lemma 3.9** (Self-Bounding, Simulation Lemma). *For any state $x$, non-negative cost function $c$, and model $W$, we have:*

$$J^\pi(x; c, W^\star) - J^\pi(x; c, W) \leq \sqrt{H V^\pi(x; c, W^\star)} \sqrt{\mathbb{E}\left[ \sum_{h=0}^{H-1} \min \left\{ \frac{\|(W^\star - W)\,\phi(x_h, u_h)\|_2^2}{\sigma^2}, 1 \right\} \right]}$$

*where the expectation is with respect to $\pi$ in $W^\star$ starting at $x_0 = x$.*

The proof, provided in Appendix B, involves the construction of a certain stopping time martingale to handle the unbounded nature of the (realized) cumulative costs, along with a novel way to handle Gaussian smoothing through the chi-squared distance function between two distributions. This lemma helps us in constructing a potential function for the analysis of the LC³ algorithm.

Table 1: Final performances for six Gym environments. Algorithm are run under the same conditions of Wang et al. [2019]. The performances of PETS-CEM and PILCO are copied for reference, and the performance of ground-truth MPPI (GT-MPPI) that has access to true model are also shown. The results are averaged over four random seeds and a window size of 5,000 timesteps.

|  | InvertedPendulum | Acrobot | CartPole | Mountain Car | Reacher | Hopper |
|---|---|---|---|---|---|---|
| $LC^3$ | $-0.0 \pm 0.0$ | $95.4 \pm 52.5$ | $199.7 \pm 0.4$ | $27.3 \pm 8.1$ | $-4.1 \pm 1.6$ | $-1016.5 \pm 607.4$ |
| (Ranking) | 1/11 | 1/11 | 2/11 | 2/11 | 1/11 | 7/11 |
| GT-MPPI | $-0.0 \pm 0.0$ | $177.8 \pm 25.0$ | $199.8 \pm 0.1$ | $24.9 \pm 2.9$ | $-2.4 \pm 0.1$ | $2995.7 \pm 215.3$ |
| PETS-CEM | $-20.5 \pm 28.9$ | $12.5 \pm 29.0$ | $199.5 \pm 3.0$ | $-57.9 \pm 3.6$ | $-12.3 \pm 5.2$ | $1125.0 \pm 679.6$ |
| PILCO | $-194.5 \pm 0.8$ | $-394.4 \pm 1.4$ | $-1.9 \pm 155.9$ | $-59.0 \pm 4.6$ | $-13.2 \pm 5.9$ | $-1729.9 \pm 1611.1$ |

## 4 Experiments

We evaluate $LC^3$ on three domains: a set of continuous control tasks, a maze environment that requires exploration, and a dexterous manipulation task. Throughout these experiments, we use model predictive path integral control (MPPI) [Williams et al., 2017] for planning, and posterior sampling [Chapelle and Li, 2011, Russo and Van Roy, 2014] for exploration– we don't implement the optimism version of $LC^3$ as analyzed, but rather implement a Thompson sampling variation. A Bayesian regret of TS is plausible using the framework developed from Russo and Van Roy [2014]. The algorithms are implemented in the Lyceum framework under the Julia programming language [Summers et al., 2020, Bezanson et al., 2017]. Comparison algorithms provided by Wang and Ba [2019], Wang et al. [2019]. Note that these experiments use reward (negative cost) for evaluations. Further details of the experiments in this section can be found in Appendix D.

**Benchmark Tasks with Random Features**  We use some common benchmark tasks, including MuJoCo [Todorov et al., 2012] environments from OpenAI Gym [Brockman et al., 2016]. We use Random Fourier Features (RFF) [Rahimi and Recht, 2008] to represent $\phi$. Table 1 shows the final performances (at 200k timesteps) of $LC^3$ with RFFs for six environments, and includes its ranking compared to the benchmarks results from Wang et al. [2019]. We find that $LC^3$ consistently performs well on simple continuous control tasks, and it works well even without posterior sampling. When the dynamical complexity increases, such as with the contact-rich Hopper model, our method's performance suffers, suggesting that these scenarios require different feature representation.

**Exploring the Maze**  We construct a maze environment to study the exploration capability of $LC^3$ (see Fig. 1 (Left)). State and control take values in $[-1,1]^2 \subset \mathbb{R}^2$ and in $[-1,1] \subset \mathbb{R}$, respectively. The task is to bring an agent to the goal state being guided by the negative cost (reward) $-c(x_h, u_h) = 8 - \|x_h - [1,1]^\top\|_2^2$. We use a one-hot vector of states and actions as features. We compare the performances of $LC^3$ to random walk that takes actions uniformly sampled within $[-1,1]$, and to PETS-CEM, which is a representative model-based RL which uses uncertainty of dynamics but without exploration. Fig. 1 (Right) plots the number of state-action pairs visited over episodes. The agent always reaches the goal within 50 episodes under the best setting of $LC^3$ while random walk and PETS-CEM never be successful within 50 episodes.

**Practical Application**  As we might consider learning model dynamics for the real world in applications such as robotics, we need sufficiently complex features. One solution to this problem is creating an ensemble of parametric models, such as found in Tobin et al. [2017], Mordatch et al. [2015]. In this experiment, we simulate a 33 degree of freedom robotic arm and hand system (see Fig. 2 (Left)) that is tasked with picking up an spherical object, and we test $LC^3$ with features $\phi$ created by six ensemble of MuJoCo models. Fig. 2 (Right) plots the learning curves of $LC^3$ with different features.

## 5 Discussion

This work considers Kernelized Nonlinear Regulators (KNRs) and provides $O(\sqrt{T})$ regret bounds for KNRs in finite horizon episodic setting, where we utilize a number of analysis concepts from reinforcement learning and machine learning for continuous problems. Our analysis works for infinite

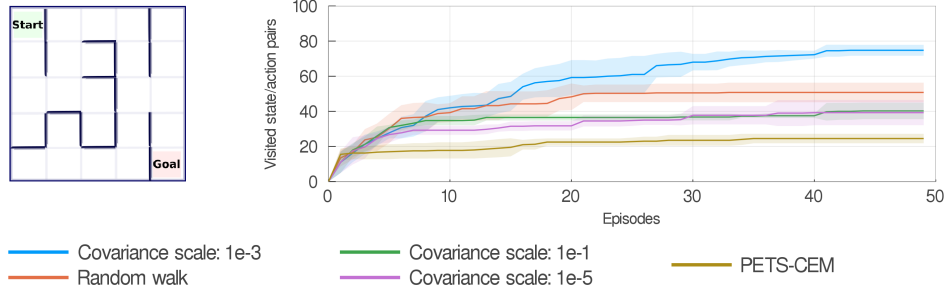

Figure 1: Left: An illustration of the maze environment. Start and End states are $[-1, -1]^\top$ and $[1, 1]^\top$, respectively. Dark lines are "walls". Right: The means and standard deviations, across four random seeds, of the number of state-action pairs already explored over episodes. Covariance scale is the posterior reshaping constant of Thompson sampling.

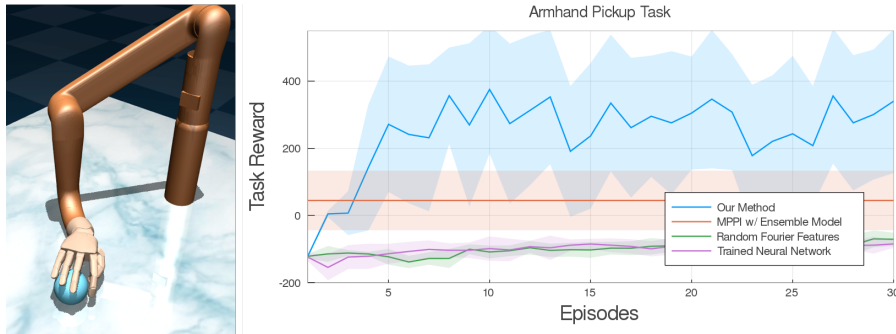

Figure 2: Left: An illustration of Armhand environment. Right: Performance curves, across 12 random seeds, of 1) LC$^3$ with ensemble features of six parametric models, 2) RFF features, and 3) the top layers of a neural network trained on a data set of 30 optimized trajectories with the correct model. Within 10 attempts at the task, LC$^3$ with ensemble features is successful. For reference, we include the average reward of MPPI using a random model from the ensemble.

dimensional feature space, unbounded state space, and unbounded cost function. The key assumption we use in our analysis is the bounded second moment of total cost (Assumption 2), which is related to the stability of the system under the policies in the policy class.

Besides, we list a number of important future directions below.

**Lower bounds:**  Sharp lower bounds would be important to develop for this very natural model. As discussed in Remark 3.5, our results are already minimax optimal for some parameter scalings.

**Improved upper bounds & $J^\star$ vs $\sqrt{J^\star}$ dependencies:**  We conjecture with stronger assumptions on higher order moments that an optimal $O(\sqrt{H^2 T})$ regret is achievable. It is also plausible that with further higher moment assumptions then, for the stabilizing case, the dependence on $J^\star$ can be improved to $\sqrt{J^\star}$. Here, our conjecture is that one would, instead, need to make a boundedness assumption on the "index of dispersion," i.e., that the ratio of the variance to the mean is bounded; we currently assume the ratio of the standard deviation to the mean is bounded.

**The single trajectory case:**  It would be interesting to use these techniques to develop regret bounds for the single trajectory case, under stronger stability and mixing assumptions of the process (see Cohen et al. [2019] for the LQR case).

**Feature learning:**  As of now, we have assumed the RKHS is known. A practically relevant direction would be to learn a good feature space.

# 6 Broader Impact

In this work, we present the first provably efficient algorithm for learning to control for Kernelized Nonlinear Regulator which is originally proposed in control literature as Gaussian Process model. Though our work focuses on the theoretical foundations of learning in nonlinear control, we believe our work has broader impact in the following aspects.

Our work connects two communities: Reinforcement Learning Theory and Control Theory. Existing models considered in RL literature that have provable guarantees hardly capture any continuous control problems while existing control theory does not focus on the sample complexity aspect of controlling unknown dynamical systems. Our work, for the first time, demonstrates that the popular KNR model from control literature, is learnable from a learning theoretical perspective. While it seems that two communities are more separated than would be ideal, we believe our work paves a new way for further communication between two communities.

From a practical application perspective, the sample efficiency of our algorithm enables control in complex dynamical settings without onerous large scale data collection, hence demonstrates potentials for model learning and control in real world applications such as dexterous manipulation, medical robotics, human robot interaction, and self-driving cars where complicated nonlinear dynamics are involved and data is often extremely expensive to collect.

Also, this line of work may be helpful to provide new means for handling model uncertainty in nonlinear planning, which may be relevant in the broader context of safety and reliability. A clear caveat here is understanding the role of model misspecification.

Lastly, but probably most importantly, our proposed algorithm along with other reinforcement learning algorithms could be ethically or physically harmful if misused; one needs to be very careful about if the assumptions we made in this paper are reasonable in the application domains of interest and if the cost function design is technically/ethically validated.

## Acknowledgments and Disclosure of Funding

The authors thank anonymous reviewers for their careful reviews and constructive comments. Also, the authors wish to thank Horia Mania for graciously sharing his thoughts in this line of work. Motoya Ohnishi thanks Aravind Rajeswaran, Vikash Kumar, and Ben Evans at Movement Control Laboratory for valuable discussions on model-based RL. He also thanks Colin Summers for instructions on Lyceum. Kendall Lowrey and Motoya Ohnishi thank Emanuel Todorov for valuable discussions and Roboti LLC for computational supports. Part of the work was done when Wen Sun was at Microsoft Research NYC. Sham Kakade gratefully acknowledges funding from the ONR award N00014-18-1-2247, and NSF Awards CCF-1703574 and CCF-1740551. Motoya Ohnishi was supported in part by Wissner-Slivka Endowed Fellowship and Funai Overseas Scholarship.

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
