[Supplementary Material]

## A  Additional Notation

A controller is a mapping $\pi : \mathcal{X} \times \{0, \ldots H-1\} \to \mathcal{U}$. Given a instantaneous cost function $c : \mathcal{X} \times \mathcal{U} \to \mathbb{R}$, we define the cost (or the "cost-to-go" ) of a policy as:

$$J^\pi(x; c, W) = \mathbb{E}\left[ \sum_{h=0}^{H-1} c(x_h, u_h) \Big| \pi, x_0 = x, W \right]$$

where the expectation is under trajectories sampled under $\pi$ starting from $x_0$ in model parameterized by $W$. The "cost-to-go" at state $x$ at time $h \in \{0, \ldots H-1\}$ is denoted by:

$$J_h^\pi(x; c, W) = \mathbb{E}\left[ \sum_{\ell=h}^{H-1} c(x_\ell, u_\ell) \Big| \pi, x_h = x \right].$$

When clear from context, we let the episode $t$ index the policy, e.g. we write $J^t(x; c)$ to refer to $J^{\pi^t}(x, c)$. Subscripts refer to the timestep within an episode and superscripts index the episode itself, i.e. $\phi_h^t$ will refer to the random vector which is the observed features during timestep $h$ within episode $t$. We let $\mathcal{H}_t$ denote the history up to the beginning of episode $t$.

Also, $\|x\|_M^2 := x^\top M x$ for a vector $x$ and a matrix $M$.

## B  Lower Confidence Bound based Analysis

In this section, we provide proofs for the two main theorems: Theorem 3.2 and Theorem 3.8.

### B.1  Simulation Analysis

We derive a novel self-bounding simulation lemma (Lemma B.3) in this section, using the Optional Stopping Theorem.

**Lemma B.1** (Difference Lemma). *Fix a policy $\pi$, cost function $c$, and model $W$. Consider any trajectory $\{x_h, u_h\}_{h=0}^{H-1}$ where $u_h = \pi(x_h)$ for all $h \in \{0, \ldots H-1\}$. For $h \in \{0, \ldots H-1\}$, let $\widehat{J}_h$ refer to the realized cost-to-go on this trajectory, i.e.*

$$\widehat{J}_h = \sum_{\tau=h}^{H-1} c(x_\tau, u_\tau).$$

*For all $\tau \in \{1, \ldots H-1\}$, we have that:*

$$\widehat{J}_0 - J_0^\pi(x_0; c, W) = \widehat{J}_\tau - \mathbb{E}_{x'_\tau \sim P(\cdot|W, x_{\tau-1}, u_{\tau-1})} J_\tau^\pi(x'_\tau; c, W)$$

$$+ \sum_{h=1}^{\tau-1} J_h^\pi(x_h; c, W) - \mathbb{E}_{x'_h \sim P(\cdot|W, x_{h-1}, u_{h-1})} J_h^\pi(x'_h; c, W)$$

*Proof.* Starting from $h = 0$, using $u_0 = \pi(x_0)$, we have:

$$\widehat{J}_0 - J_0^\pi(x_0; c, W) = \widehat{J}_1 - \mathbb{E}_{x'_1 \sim P(\cdot|W, x_0, u_0)} J_1^\pi(x'_1; c, W)$$

$$= \widehat{J}_1 - J_1^\pi(x_1; c, W) + J_1^\pi(x_1; c, W) - \mathbb{E}_{x'_1 \sim P(\cdot|W, x_0, u_0)} J_\ell^\pi(x'_1; c, W)$$

$$= \widehat{J}_2 - \mathbb{E}_{x'_2 \sim P(\cdot|x_1, u_1, W)} J_2^\pi(x'_2; c, W)$$

$$+ J_1^\pi(x_1; c, W) - \mathbb{E}_{x'_1 \sim P(\cdot|W, x_0, u_0)} J_1^\pi(x'_1; c, W).$$

Recursion completes the proof, where, at each step of the recursion, we add and subtract $J_t^\pi(x_t; c, W)$ and apply the same operation on the term $\widehat{J}_t - J_t^\pi(x_t; c, W)$;  □

**Lemma B.2** ("Optional Stopping" Simulation Lemma). *Fix a policy $\pi$, cost function $c$, and model $W$. Consider the stochastic process over trajectories, where $\{x_h, u_h\}_{h=0}^{H} \sim \pi$ is sampled with respect to the model $W^\star$. With respect to this stochastic process, define a stopping time $\tau$ as:*

$$\tau = \min \{h \geq 0 : J_h^\pi(x_h; c, W) \geq J_h^\pi(x_h; c, W^\star)\}.$$

*Define the random variable $\widetilde{J}_h^\pi(x_h)$ as:*

$$\widetilde{J}_h^\pi(x_h) = \min\left\{J_h^\pi(x_h; c, W), J_h^\pi(x_h; c, W^\star)\right\}.$$

*We have that:*

$$J_0^\pi(x_0; c, W^\star) - J_0^\pi(x_0; c, W)$$

$$\leq \mathbb{E}\left[\sum_{h=0}^{H-1} 1\{h < \tau\}\left(\mathbb{E}_{x'_{h+1} \sim P(\cdot|W^\star, x_h, u_h)}\widetilde{J}_{h+1}^\pi(x'_{h+1}) - \mathbb{E}_{x'_{h+1} \sim P(\cdot|W, x_h, u_h)}\widetilde{J}_{h+1}^\pi(x'_{h+1})\right)\right]$$

*where the expectation is with respect to $\{x_h, u_h\}_{h=0}^{H} \sim \pi$ sampled with respect to the model $W^\star$.*

*Proof.* Our filtration, $\mathcal{F}_h$, at time $h$ will be the previous noise variables, i.e.

$$\mathcal{F}_h := \{\epsilon_0, \epsilon_1 \ldots \epsilon_{h-1}\},$$

and note that $\{x_1, u_1, c(x_1, u_1), \ldots x_h, u_h, c(x_h, u_h)\}$ is fully determined by $\mathcal{F}_h$. Also, observe that $\tau$ is a valid stopping time with respect to the filtration $\mathcal{F}_h$.

Define:

$$M_h = \mathbb{E}\left[\widehat{J}_0 - J^\star(x_0; c, W) \mid \mathcal{F}_h\right]$$

which is a Doob martingale (with respect to our filtration), and so $\mathbb{E}[M_{h+1}|\mathcal{F}_h] = M_h$. By Doob's optional stopping theorem,

$$\mathbb{E}\left[\widehat{J}_0 - J^\star(x_0; c, W)\right] = \mathbb{E}[M_\tau] = \mathbb{E}\left[\mathbb{E}\left[\widehat{J}_0 - J^\star(x_0; c, W) \mid \mathcal{F}_\tau\right]\right]. \tag{B.1}$$

The proof consists in bounding $M_\tau$.

Consider an $\mathcal{F}_\tau$, which is stopped at the random time $\tau$. By Lemma B.1,

$$
\begin{aligned}
M_\tau &= \mathbb{E}\left[\widehat{J}_0 - J^\star(x_0; c, W) \mid \mathcal{F}_\tau\right]\\
&= J_\tau(x_\tau; c, W^\star) - \mathbb{E}_{x'_\tau \sim P(\cdot|W, x_{\tau-1}, u_{\tau-1})}J_h(x'_\tau; c, W)\\
&\quad + \sum_{h=1}^{\tau-1}\left(J_h(x_h; c, W) - \mathbb{E}_{x'_h \sim P(\cdot|W, x_{h-1}, u_{h-1})}J_h(x'_h; c, W)\right)\\
&= \sum_{h=1}^{\tau}\left(\widetilde{J}_h(x_h) - \mathbb{E}_{x'_h \sim P(\cdot|W, x_{h-1}, u_{h-1})}J_h(x'_h; c, W)\right)\\
&\leq \sum_{h=1}^{\tau}\left(\widetilde{J}_h(x_h) - \mathbb{E}_{x'_h \sim P(\cdot|W, x_{h-1}, u_{h-1})}\widetilde{J}_h(x'_h)\right)\\
&= \sum_{h=1}^{H} 1(h \leq \tau)\left(\widetilde{J}_h(x_h) - \mathbb{E}_{x'_h \sim P(\cdot|W, x_{h-1}, u_{h-1})}\widetilde{J}_h(x'_h)\right).
\end{aligned}
$$

where the third equality follows using the definition of $\tau$ which implies that $J_\tau(x_\tau; c, W^\star) = \widetilde{J}_\tau(x_\tau)$ and that $J_h(x_h; c, W) = \widetilde{J}_h(x_h)$ for $h < \tau$; and the inequality is due to the definition of $\widetilde{J}$.

Using this bound on $M_\tau$ and Equation B.1, we have:

$$\mathbb{E}\left[\widehat{J}_0 - J^\star(x_0; c, W)\right] \leq \sum_{h=1}^{H} \mathbb{E}\left[1(h \leq \tau)\left(\widetilde{J}_h(x_h) - \mathbb{E}_{x'_h \sim P(\cdot|W, x_{h-1}, u_{h-1})}\widetilde{J}_h(x'_h)\right)\right].$$

For the $h$-th term, observe:

$$
\begin{aligned}
&\mathbb{E}\left[1(h \leq \tau)\left(\widetilde{J}_h(x_h) - \mathbb{E}_{x'_h \sim P(\cdot|W, x_{h-1}, u_{h-1})}\widetilde{J}_h(x'_h)\right)\right]\\
&= \mathbb{E}\left[\mathbb{E}\left[1(h \leq \tau)\left(\widetilde{J}_h(x_h) - \mathbb{E}_{x'_h \sim P(\cdot|W, x_{h-1}, u_{h-1})}\widetilde{J}_h(x'_h)\right) \mid \mathcal{F}_{h-1}\right]\right]\\
&= \mathbb{E}\left[\mathbb{E}\left[1(h-1 < \tau)\left(\widetilde{J}_h(x_h) - \mathbb{E}_{x'_h \sim P(\cdot|W, x_{h-1}, u_{h-1})}\widetilde{J}_h(x'_h)\right) \mid \mathcal{F}_{h-1}\right]\right]\\
&= \mathbb{E}\left[1(h-1 < \tau)\mathbb{E}\left[\widetilde{J}_h(x_h) - \mathbb{E}_{x'_h \sim P(\cdot|W, x_{h-1}, u_{h-1})}\widetilde{J}_h(x'_h) \mid \mathcal{F}_{h-1}\right]\right]\\
&= \mathbb{E}\left[1(h-1 < \tau)\left(\mathbb{E}_{x'_h \sim P(\cdot|W^\star, x_{h-1}, u_{h-1})}\widetilde{J}_h(x'_h) - \mathbb{E}_{x'_h \sim P(\cdot|W, x_{h-1}, u_{h-1})}\widetilde{J}_h(x'_h)\right)\right].
\end{aligned}
$$

where the second equality uses that $1(h \leq \tau) = 1(h - 1 < \tau)$, and the third equality uses that $1(h - 1 < \tau)$ is measurable with respect to $\mathcal{F}_{h-1} = \{\epsilon_0, \dots \epsilon_{h-2}\}$. This completes the proof. $\qquad \square$

The previous lemma allows us to bound the difference in cost under two different models, i.e. $J^\pi(x; c, W^\star) - J^\pi(x; c, W)$, in terms of the second moment of the cumulative cost itself, i.e. in terms of $V^\pi(x; c, W^\star)$, where

$$V^\pi(x_0; c, W^\star) := \mathbb{E}\left[\left(\sum_{h=0}^{H-1} c(x_h, u_h)\right)^2 \mid x_0, \pi, W^\star\right].$$

**Lemma B.3** (Self-Bounding, Simulation Lemma). *For any policy $\pi$, model parameterization $W$, and non-negative cost $c$, and for any state $x_0$, we have:*

$$J^\pi(x_0; c, W^\star) - J^\pi(x_0; c, W)$$

$$\leq \sqrt{HV^\pi(x_0; c, W^\star)}\sqrt{\mathbb{E}\left[\sum_{h=0}^{H-1} \min\left\{\frac{1}{\sigma^2} \|(W^\star - W)\phi(x_h, u_h)\|_2^2, 1\right\}\right]}.$$

*where the expectation is with respect to $\pi$ in $W^\star$ starting at $x_0$.*

*Proof.* For the proof, it is helpful to define the random variables:

$$\Delta_h \;=\; \mathbb{E}_{x'_{h+1} \sim P(\cdot|W^\star, x_h, u_h)}\left[\widetilde{J}_{h+1}(x'_{h+1})\right] - \mathbb{E}_{x'_{h+1} \sim P(\cdot|W, x_h, u_h)}\left[\widetilde{J}_{h+1}(x'_{h+1})\right]$$

$$A_h \;:=\; \mathbb{E}_{x'_{h+1} \sim P(\cdot|W^\star, x_h, u_h)}\left[\widetilde{J}_{h+1}(x'_{h+1})^2\right]$$

By Lemma C.2 (which bounds the difference in means under two Gaussian distributions, using the chi-squared distance function), we have:

$$\Delta_h \leq \sqrt{\mathbb{E}_{x_{h+1} \sim P(\cdot|W^\star, x_h, u_h)}\left[\widetilde{J}_{h+1}(x_{h+1})^2\right]} \min\left\{\frac{1}{\sigma} \|(W^\star - W)\phi(x_h, u_h)\|_2, 1\right\}$$

$$= \sqrt{A_h} \min\left\{\frac{1}{\sigma} \|(W^\star - W)\phi(x_h, u_h)\|_2, 1\right\}.$$

From Lemma B.2, we have:

$$J_0^\pi(x_0; c, W^\star) - J_0^\pi(x_0; c, W) \leq \sum_{h=0}^{H-1} \mathbb{E}\left[1(h < \tau)\Delta_h\right]$$

$$\leq \sum_{h=0}^{H-1} \mathbb{E}\left[\sqrt{A_h} \min\left\{\frac{1}{\sigma} \|(W^\star - W)\phi(x_h, u_h)\|_2, 1\right\}\right]$$

$$\leq \sum_{h=0}^{H-1} \sqrt{\mathbb{E}\left[A_h\right]}\sqrt{\mathbb{E}\left[\min\left\{\frac{1}{\sigma^2} \|(W^\star - W)\phi(x_h, u_h)\|_2^2, 1\right\}\right]}$$

$$\leq \sqrt{\mathbb{E}\left[\sum_{h=0}^{H-1} A_h\right]}\sqrt{\mathbb{E}\left[\sum_{h=0}^{H-1} \min\left\{\frac{1}{\sigma^2} \|(W^\star - W)\phi(x_h, u_h)\|_2^2, 1\right\}\right]},$$

where in the second inequality we use $\mathbb{E}[ab] \leq \sqrt{\mathbb{E}[a^2]\mathbb{E}[b^2]}$ and the Cauchy-Schwartz inequality in the last inequality. For the first term, observe that:

$$\mathbb{E}\left[A_h\right] = \mathbb{E}\left[\mathbb{E}_{x'_{h+1} \sim P(\cdot|W^\star, x_h, u_h)}\left[\widetilde{J}_{h+1}(x'_{h+1})^2\right]\right] = \mathbb{E}\left[\widetilde{J}_{h+1}(x_{h+1})^2\right] \leq \mathbb{E}\left[J_{h+1}(x_{h+1})^2\right]$$

$$= \mathbb{E}\left[\left(\mathbb{E}\left[\sum_{\ell=h+1}^{H-1} c(x_\ell, u_\ell) \mid x_{h+1}\right]\right)^2\right] \leq \mathbb{E}\left[\left(\sum_{\ell=h+1}^{H-1} c(x_\ell, u_\ell)\right)^2\right]$$

$$\leq \mathbb{E}\left[\left(\sum_{\ell=0}^{H-1} c(x_\ell, u_\ell)\right)^2\right] = V^\pi$$

where the first inequality uses the definition of $\widetilde{J}$; the second inequality follows from Jensen's inequality; and the last inequality follows from our assumption that the instantaneous costs are non-negative. The proof is completed by substitution. $\qquad\square$

## B.2 Regret Analysis (and proofs of Theorem 3.2 and Theorem 3.8)

Throughout, let $\mathcal{E}_{t,cb}$ be the event that $W^\star \in \mathrm{BALL}^t$ holds at episode $t$.

**Lemma B.4** (Per-episode Regret Lemma)**.** *Suppose Assumptions 1 and 2 hold. Let $\mathcal{H}_{<t}$ be the history of events before episode $t$. For the LC$^3$, we have:*

$$\mathbf{1}(\mathcal{E}_{t,cb})\Big(J^t(x_0;c^t) - J^\star(x_0;c^t)\Big)$$

$$\leq \sqrt{HV^t(x_0;c,W^\star)\left(\frac{4\beta^t}{\sigma^2}+H\right)}\sqrt{\mathbb{E}\left[\min\left\{\sum_{h=0}^{H-1}\|\phi_h^t\|_{(\Sigma^t)^{-1}}^2,1\right\}\;\Big|\;\mathcal{H}_{<t}\right]}.$$

*Note that the expectation is with respect to the trajectory of LC$^3$, i.e. it is under $\pi^t$ in $W^\star$.*

*Proof.* Suppose $\mathcal{E}_{cb}^t$ holds, else the lemma is immediate. By construction of the LC$^3$ algorithm (the optimistic property) and by the self-bounding, simulation lemma (Lemma B.3), we have:

$$J^t(x_0;c^t,W^\star) - J^\star(x_0;c^t,W^\star) \leq J^t(x_0;c^t,W^\star) - J^t(x_0;c^t,\widehat{W}^t)$$

$$\leq \sqrt{HV^t(x_0;c,W^\star)}\sqrt{\mathbb{E}\left[\sum_{h=0}^{H-1}\min\left\{\frac{1}{\sigma^2}\left\|\left(W^\star - \widehat{W}^t\right)\phi_h^t\right\|_2^2,1\right\}\;\Big|\;\mathcal{H}_{<t}\right]}.$$

where the expectation is with respect to the trajectory of LC$^3$, i.e. of $\pi^t$ in $W^\star$.

For $W^\star \in \mathrm{BALL}^t$, we have

$$\left\|\left(\widehat{W}^t - W^\star\right)\phi_h^t\right\|_2 \leq \left\|\left(\widehat{W}^t - W^\star\right)(\Sigma^t)^{1/2}\right\|_2\left\|(\Sigma^t)^{-1/2}\phi_h^t\right\|_2$$

$$\leq \left(\left\|\left(\widehat{W}^t - \overline{W}^t\right)(\Sigma^t)^{1/2}\right\|_2 + \left\|\left(\overline{W}^t - W^\star\right)(\Sigma^t)^{1/2}\right\|_2\right)\|\phi_h^t\|_{(\Sigma^t)^{-1}} \leq 2\sqrt{\beta^t}\|\phi_h^t\|_{(\Sigma^t)^{-1}}.$$

where we have also used that $\widehat{W}^t, \overline{W}^t \in \mathrm{BALL}^t$, by construction.

This implies that:

$$\sum_{h=0}^{H-1}\min\left\{\frac{1}{\sigma^2}\|(W^\star - \widehat{W}^t)\phi_h^t\|_2^2,1\right\} \leq \sum_{h=0}^{H-1}\min\left\{\frac{4\beta^t}{\sigma^2}\|\phi_h^t\|_{(\Sigma^t)^{-1}}^2,1\right\}$$

$$\leq \min\left\{\frac{4\beta^t}{\sigma^2}\sum_{h=0}^{H-1}\|\phi_h^t\|_{(\Sigma^t)^{-1}}^2,H\right\} \leq \max\left\{\frac{4\beta^t}{\sigma^2},H\right\}\min\left\{\sum_{h=0}^{H-1}\|\phi_h^t\|_{(\Sigma^t)^{-1}}^2,1\right\}.$$

The proof is completed by substitution. $\qquad\square$

Before we complete the proofs, the following two lemmas are helpful. Their proofs are provided in Appendix B.3. The first lemma bounds the sum failure probability of $W^\star$ not being in all the confidence balls (over all the episodes); the lemma generalizes the argument from [Abbasi-Yadkori et al., 2011, Dani et al., 2008] to matrix regression.

**Lemma B.5** (Confidence Ball)**.** *Let*

$$\beta^t = 2\lambda\|W^\star\|_2^2 + 8\sigma^2\left(d_\mathcal{X}\log(5) + 2\log(t) + \log(4) + \log\left(\det(\Sigma^t)/\det(\Sigma^0)\right)\right).$$

*We have:*

$$\sum_{t=0}^{\infty}\Pr\left(\overline{\mathcal{E}}_{t,cb}\right) = \sum_{t=0}^{\infty}\Pr\left(\left\|\left(\overline{W}^t - W^\star\right)(\Sigma^t)^{1/2}\right\|_2^2 > \beta^t\right) \leq \frac{1}{2}.$$

The next lemma provides a bound on the potential function used in our analysis. It is based on the elliptical potential function argument from [Dani et al., 2008, Srinivas et al., 2009].

**Lemma B.6** (Sum of Potential Functions). *For any sequence of $\phi_h^t$, we have:*

$$\sum_{t=0}^{T-1} \min \left\{ \sum_{h=0}^{H-1} \|\phi_h^t\|_{(\Sigma^t)^{-1}}^2, 1 \right\} \le 2 \log \left( \det(\Sigma^T) \det(\Sigma^0)^{-1} \right).$$

Also, recall that LC$^3$ uses the setting of $\lambda = \sigma^2 / \|W^\star\|_2^2$. We will also use that, for $\beta^T$ as defined in Lemma B.5,

$$\beta^T = 2\sigma^2 + 8\sigma^2 \left( d_{\mathcal{X}} \log(5) + 2\log(T) + \log(4) + \log \left( \det(\Sigma^T) \det(\Sigma^0)^{-1} \right) \right)$$
$$\le 16\sigma^2 \left( d_{\mathcal{X}} + \log(T) + \log \left( \det(\Sigma^T) \det(\Sigma^0)^{-1} \right) \right). \tag{B.2}$$

In particular, we can take $C_1 = 16$ in LC$^3$. Also,

$$\mathbb{E}[\beta^T] \le 16\sigma^2 \left( d_{\mathcal{X}} + \log(T) + \gamma_T(\lambda) \right). \tag{B.3}$$

using the definition of the information gain.

We now conclude the proof our first main theorem (Theorem 3.2).

*Proof of Theorem 3.2.* Using the per-episode regret bound (Lemma B.4), our confidence ball, failure probability bound (Lemma B.5), and that $V^t \le V_{\max}$,

$$\mathbb{E}\left[ \text{REGRET}_{LC3} \right] = \mathbb{E}\left[ \sum_{t=0}^{T-1} \left( J^t(x_0; c^t) - J^\star(x_0; c^t) \right) \right]$$

$$\le \mathbb{E}\left[ \sum_{t=0}^{T-1} \mathbb{E}\left[ \mathbf{1}(\mathcal{E}_{t,cb}) \left( J^t(x_0; c^t) - J^\star(x_0; c^t) \right) \mid \mathcal{H}_t \right] \right] + \sqrt{V_{\max}} \sum_{t=0}^{T-1} \mathbb{E}\left[ \mathbf{1}(\overline{\mathcal{E}}_{t,cb}) \right]$$

$$\le \sqrt{H V_{\max}} \sum_{t=0}^{T-1} \mathbb{E}\left[ \sqrt{\frac{4\beta^t}{\sigma^2} + H} \sqrt{\mathbb{E}\left[ \min \left\{ \sum_{h=0}^{H-1} \|\phi_h^t\|_{(\Sigma^t)^{-1}}^2, 1 \right\} \mid \mathcal{H}_{<t} \right]} \right] + \sqrt{V_{\max}}/2$$

$$\le \sqrt{H V_{\max}} \sum_{t=0}^{T-1} \sqrt{\mathbb{E}\left[ \frac{4\beta^t}{\sigma^2} + H \right]} \sqrt{\mathbb{E}\left[ \min \left\{ \sum_{h=0}^{H-1} \|\phi_h^t\|_{(\Sigma^t)^{-1}}^2, 1 \right\} \right]} + \sqrt{V_{\max}}/2$$

$$\le \sqrt{H V_{\max}} \sqrt{\sum_{t=0}^{T-1} \mathbb{E}\left[ \frac{4\beta^t}{\sigma^2} + H \right]} \sqrt{\mathbb{E}\left[ \sum_{t=0}^{T-1} \min \left\{ \sum_{h=0}^{H-1} \|\phi_h^t\|_{(\Sigma^t)^{-1}}^2, 1 \right\} \right]} + \sqrt{V_{\max}}/2$$

$$\le \sqrt{H V_{\max}} \sqrt{T \left( \frac{4\mathbb{E}[\beta^T]}{\sigma^2} + H \right)} \sqrt{\gamma_T(\lambda)} + \sqrt{V_{\max}}/2$$

$$\le \sqrt{H V_{\max}} \sqrt{64 T \left( d_{\mathcal{X}} + \log(T) + \gamma_T(\lambda) + H \right)} \sqrt{\gamma_T(\lambda)} + \sqrt{V_{\max}}/2$$

where the third inequality use that $\mathbb{E}[ab] \le \sqrt{\mathbb{E}[a^2]\mathbb{E}[b^2]}$; the fourth uses the Cauchy-Schwartz inequality; the penultimate step uses that $\beta_t$ is non-decreasing, along with the Lemma B.6 and the definition of the information gain; and the final step uses the bound on $\beta^T$ in Equation B.3. This completes the proof. $\qquad \square$

The proof of our second main theorem (Theorem 3.8) now follows.

*Proof of Theorem 3.8.* By assumption 3 on $V^t$ and the per-episode regret lemma (Lemma B.4),

$$\mathbf{1}(\mathcal{E}_{t,cb}) V^t \le \alpha^2 \mathbf{1}(\mathcal{E}_{t,cb}) J^t(x_0; c^t, W^\star)^2$$

$$\le 2\alpha^2 J^\star(x_0; c^t, W^\star)^2 + 2\mathbf{1}(\mathcal{E}_{t,cb}) \alpha^2 \left( J^t(x_0; c^t, W^\star) - J^\star(x_0; c^t, W^\star) \right)^2$$

$$\le 2\alpha^2 J^\star(x_0; c^t, W^\star)^2 + 2\alpha^2 H V_{\max} \left( \frac{4\beta^t}{\sigma^2} + H \right) \mathbb{E}\left[ \min \left\{ \sum_{h=0}^{H-1} \|\phi_h^t\|_{(\Sigma^t)^{-1}}^2, 1 \right\} \mid \mathcal{H}_{<t} \right]$$

Using this, and with Lemma B.4 and Lemma B.5, we have

$$\mathbb{E}\left[\mathrm{REGRET}_{LC^3}\right]$$

$$\leq \mathbb{E}\left[\sum_{t=0}^{T-1}\mathbb{E}\left[\mathbf{1}(\mathcal{E}_{t,cb})\left(J^t(x_0;c^t)-J^\star(x_0;c^t)\right)\mid\mathcal{H}_t\right]\right]+\sqrt{V_{\max}}\sum_{t=0}^{T-1}\mathbb{E}\left[\mathbf{1}(\overline{\mathcal{E}}_{t,cb})\right]$$

$$\leq \sum_{t=0}^{T-1}\mathbb{E}\left[\sqrt{H\mathbf{1}(\mathcal{E}_{t,cb})V^t\left(\frac{4\beta^t}{\sigma^2}+H\right)}\sqrt{\mathbb{E}\left[\min\left\{\sum_{h=0}^{H-1}\|\phi_h^t\|_{(\Sigma^t)^{-1}}^2,1\right\}\ \Big|\ \mathcal{H}_{<t}\right]}\right]+\sqrt{V_{\max}}/2$$

$$\leq \alpha J^*\sqrt{2H}\sum_{t=0}^{T-1}\mathbb{E}\left[\sqrt{\frac{4\beta^t}{\sigma^2}+H}\sqrt{\mathbb{E}\left[\min\left\{\sum_{h=0}^{H-1}\|\phi_h^t\|_{(\Sigma^t)^{-1}}^2,1\right\}\ \Big|\ \mathcal{H}_{<t}\right]}\right]$$

$$+\ \alpha\sqrt{2H^2V_{\max}}\sum_{t=0}^{T-1}\mathbb{E}\left[\left(\frac{4\beta^t}{\sigma^2}+H\right)\mathbb{E}\left[\min\left\{\sum_{h=0}^{H-1}\|\phi_h^t\|_{(\Sigma^t)^{-1}}^2,1\right\}\ \Big|\ \mathcal{H}_{<t}\right]\right]+\sqrt{V_{\max}}/2.$$

where have used that $\sqrt{a+b}\leq\sqrt{a}+\sqrt{b}$ for positive $a$ and $b$ in the last inequality.

An identical argument to that in the proof of Theorem 3.2 leads to the first term above being bounded as:

$$\alpha J^*\sqrt{2H}\sum_{t=0}^{T-1}\mathbb{E}\left[\sqrt{\frac{4\beta^t}{\sigma^2}+H}\sqrt{\mathbb{E}\left[\min\left\{\sum_{h=0}^{H-1}\|\phi_h^t\|_{(\Sigma^t)^{-1}}^2,1\right\}\ \Big|\ \mathcal{H}_{<t}\right]}\right]$$

$$\leq \alpha J^\star\sqrt{128\gamma_T(\lambda)\left(d_{\mathcal{X}}+\log(T)+\gamma_T(\lambda)+H\right)HT}$$

For the second term,

$$\mathbb{E}\left[\sum_{t=0}^{T-1}\left(\frac{4\beta^t}{\sigma^2}+H\right)\mathbb{E}\left[\min\left\{\sum_{h=0}^{H-1}\|\phi_h^t\|_{(\Sigma^t)^{-1}}^2,1\right\}\ \Big|\ \mathcal{H}_{<t}\right]\right]$$

$$=\mathbb{E}\left[\sum_{t=0}^{T-1}\left(\frac{4\beta^t}{\sigma^2}+H\right)\min\left\{\sum_{h=0}^{H-1}\|\phi_h^t\|_{(\Sigma^t)^{-1}}^2,1\right\}\right]$$

$$\leq \mathbb{E}\left[\left(\frac{4\beta^T}{\sigma^2}+H\right)\sum_{t=0}^{T-1}\min\left\{\sum_{h=0}^{H-1}\|\phi_h^t\|_{(\Sigma^t)^{-1}}^2,1\right\}\right]$$

$$\leq 2\mathbb{E}\left[\left(\frac{4\beta^T}{\sigma^2}+H\right)\log\left(\det(\Sigma^T)\det(\Sigma^0)^{-1}\right)\right]$$

$$\leq 128\mathbb{E}\left[\left(d_{\mathcal{X}}+\log(T)+\log\left(\det(\Sigma^T)\det(\Sigma^0)^{-1}\right)+H\right)\log\left(\det(\Sigma^T)\det(\Sigma^0)^{-1}\right)\right]$$

$$\leq 128\left(H+d_{\mathcal{X}}+\log(T)\right)\gamma_T(\lambda)+128\gamma_{2,T}(\lambda)$$

where we have used that $\beta^t$ is measurable with respect to $\mathcal{H}_{<t}$ in the first equality; that $\beta^t$ is non-decreasing in the first inequality; Lemma B.6 in the second inequality; our bound on $\beta^T$ in Equation B.2 in the third inequality; and the definition of $\gamma_T(\lambda)$ and $\gamma_{2,T}(\lambda)$ in the final step.

The proof is completed via substitution. $\qquad\square$

## B.3 Confidence Bound and Potential Function Analysis

*Proof of Lemma B.5.* The center of the confidence ball, $\overline{W}^t$, is the minimizer of the ridge regression objective in Equation 3.1; its closed-form expression is:

$$\overline{W}^t := \sum_{\tau=0}^{t-1}\sum_{h=0}^{H-1}x_{h+1}^\tau(\phi_h^\tau)^\top(\Sigma^t)^{-1},$$

where $\Sigma^t = \lambda I + \sum_{\tau=0}^{t-1} \sum_{h=0}^{H-1} \phi_h^\tau (\phi_h^\tau)^\top$. Using that $x_{h+1}^\tau = W^\star \phi_h^\tau + \epsilon_h^\tau$ with $\epsilon_h^\tau \sim \mathcal{N}(0, \sigma^2 I)$,

$$\overline{W}^t - W^\star = \sum_{\tau=0}^{t-1} \sum_{h=0}^{H-1} x_{h+1}^\tau (\phi_h^\tau)^\top (\Sigma^t)^{-1} - W^\star$$

$$= \sum_{\tau=0}^{t-1} \sum_{h=0}^{H-1} (W^\star \phi_h^\tau + \epsilon_h^\tau)(\phi_h^\tau)^\top (\Sigma^t)^{-1} - W^\star$$

$$= W^\star \left( \sum_{\tau=0}^{t-1} \sum_{h=0}^{H-1} \phi_h^\tau (\phi_h^\tau)^\top \right) (\Sigma^t)^{-1} - W^\star + \sum_{\tau=0}^{t-1} \sum_{h=0}^{H-1} \epsilon_h^\tau (\phi_h^\tau)^\top (\Sigma^t)^{-1}$$

$$= -\lambda W^\star \left( \Sigma^t \right)^{-1} + \sum_{\tau=0}^{t-1} \sum_{h=0}^{H-1} \epsilon_h^\tau (\phi_h^\tau)^\top (\Sigma^t)^{-1}.$$

For any $0 < \delta_t < 1$, using Lemma C.4, it holds with probability at least $1 - \delta_t$,

$$\left\| \left( \overline{W}^t - W^\star \right) \left( \Sigma^t \right)^{1/2} \right\|_2 \leq \left\| \lambda W^\star \left( \Sigma^t \right)^{-1/2} \right\|_2 + \left\| \sum_{\tau=0}^{t-1} \sum_{h=0}^{H-1} \epsilon_h^\tau (\phi_h^\tau)^\top (\Sigma^t)^{-1/2} \right\|_2$$

$$\leq \sqrt{\lambda} \|W^\star\|_2 + \sigma \sqrt{8 d_\mathcal{X} \log(5) + 8 \log \left( \det(\Sigma^t) \det(\Sigma^0)^{-1} / \delta_t \right)}.$$

where we have also used the triangle inequality. Therefore, $\Pr(\overline{\mathcal{E}}_{t,cb}) \leq \delta_t$.

We seek to bound $\sum_{t=0}^\infty \Pr(\overline{\mathcal{E}}_{t,cb})$. Due to that at $t = 0$ we have initialized $\mathrm{BALL}^0$ to contain $W^\star$, we have $\Pr(\overline{\mathcal{E}}_{0,cb}) = 0$. For $t \geq 1$, let us assign failure probability $\delta_t = (3/\pi^2)/t^2$ for the $t$-th event, which, using the above, gives us an upper bound on the sum failure probability as $\sum_{t=1}^\infty \Pr(\overline{\mathcal{E}}_{t,cb}) < \sum_{t=1}^\infty (1/t^2)(3/\pi^2) = 1/2$. This completes the proof. $\qquad\square$

*Proof of Lemma B.6.* Recall that $\Sigma^{t+1} = \Sigma^t + \sum_{h=0}^{H-1} \phi_h^t \left( \phi_h^t \right)^\top$ and $\Sigma^0 = \lambda I$. First use $x \leq 2 \log(1 + x)$ for $x \in [0, 1]$, we have:

$$\min \left\{ \sum_{h=0}^{H-1} \|\phi_h^t\|_{(\Sigma^t)^{-1}}^2, 1 \right\} \leq 2 \log \left( 1 + \sum_{h=0}^{H-1} \|\phi_h^t\|_{(\Sigma^t)^{-1}}^2 \right).$$

For $\Sigma^{t+1}$, using its recursive formulation, we have:

$$\log \det \left( \Sigma^{t+1} \right) = \log \det \left( \Sigma^t \right) + \log \det \left( I + \left( \Sigma^t \right)^{-1/2} \sum_{h=0}^{H-1} \phi_h^t (\phi_h^t)^\top \left( \Sigma^t \right)^{-1/2} \right)$$

Denote the eigenvalues of $(\Sigma^t)^{-1/2} \sum_{h=0}^{H-1} \phi_h^t (\phi_h^t)^\top (\Sigma^t)^{-1/2}$ as $\sigma_i$ for $i \geq 1$. We have

$$\log \det \left( I + \left( \Sigma^t \right)^{-1/2} \sum_{h=0}^{H-1} \phi_h^t (\phi_h^t)^\top \left( \Sigma^t \right)^{-1/2} \right) = \log \prod_{i \geq 1} (1 + \sigma_i) \geq \log \left( 1 + \sum_{i \geq 1} \sigma_i \right),$$

where the last inequality uses that $\sigma_i \geq 0$ for all $i$. Using the above and the definition of the trace,

$$\log \det \left( I + \left( \Sigma^t \right)^{-1/2} \sum_{h=0}^{H-1} \phi_h^t (\phi_h^t)^\top \left( \Sigma^t \right)^{-1/2} \right) \geq \log \left( 1 + \mathrm{tr} \left( \left( \Sigma^t \right)^{-1/2} \sum_{h=0}^{H-1} \phi_h^t (\phi_h^t)^\top \left( \Sigma^t \right)^{-1/2} \right) \right)$$

$$= \log \left( 1 + \sum_{h=0}^{H} (\phi_h^t)^\top (\Sigma^t)^{-1} \phi_h^t \right)$$

By telescoping the sum,

$$2 \sum_{t=0}^{T-1} \log \left( 1 + \sum_{h=0}^{H} (\phi_h^t)^\top (\Sigma^t)^{-1} \phi_h^t \right) \leq 2 \sum_{t=1}^{T-1} \left( \log \det \left( \Sigma^{t+1} \right) - \log \det \left( \Sigma^t \right) \right)$$

$$= \log \left( \det(\Sigma^T) \det(\Sigma^0)^{-1} \right),$$

which completes the proof. $\qquad\square$

## C Technical Lemmas

**Lemma C.1** (Chi Squared Distance Between Two Gaussians). *For Gaussian distributions $\mathcal{N}(\mu_1, \sigma^2 I)$ and $\mathcal{N}(\mu_2, \sigma^2 I)$, the (squared) chi-squared distance between $\mathcal{N}_1$ and $\mathcal{N}_2$ is:*

$$\int \frac{(\mathcal{N}_1(z) - \mathcal{N}_2(z))^2}{\mathcal{N}_1(z)} dz = \exp\left(\frac{\|\mu_1 - \mu_2\|^2}{2\sigma^2}\right) - 1$$

*Proof.* Observe that:

$$\int \frac{(\mathcal{N}_1(z) - \mathcal{N}_2(z))^2}{\mathcal{N}_1(z)} dz = \int \mathcal{N}_1(z) - 2\mathcal{N}_2(z) + \frac{\mathcal{N}_2(z)^2}{\mathcal{N}_1(z)} dz = -1 + \int \frac{\mathcal{N}_2(z)^2}{\mathcal{N}_1(z)} dz.$$

Note that for $\mathcal{N}_2^2(z)/\mathcal{N}_1(z)$, we have:

$$\mathcal{N}_2^2(z)/\mathcal{N}_1(z) = \frac{1}{Z} \exp\left(-\frac{1}{2\sigma^2}\left(2\|z - \mu_2\|_2^2 - \|z - \mu_1\|_2^2\right)\right),$$

where $Z$ is the normalization constant for $\mathcal{N}(0, \sigma^2 I)$, i.e. $Z = \int \exp\left(-\frac{1}{2\sigma^2}\|z\|_2^2\right) dz$.

For $2\|z - \mu_2\|_2^2 - \|z - \mu_1\|_2^2$, we can verify that:

$$2\|z - \mu_2\|_2^2 - \|z - \mu_1\|_2^2 = \|z + (\mu_1 - 2\mu_2)\|_2^2 - 2\|\mu_1 - \mu_2\|_2^2.$$

This implies that:

$$\begin{aligned}
\int \frac{\mathcal{N}_2(z)^2}{\mathcal{N}_1(z)} dz &= \frac{1}{Z} \int \exp\left(-\frac{1}{2\sigma^2}\left(\|z - (2\mu_2 - \mu_1)\|_2^2 - 2\|\mu_1 - \mu_2\|\right)\right) dz \\
&= \frac{1}{Z} \exp\left(\frac{\|\mu_1 - \mu_2\|_2^2}{\sigma^2}\right) \int \exp\left(-\frac{1}{2\sigma^2}\|z - (2\mu_2 - \mu_1)\|_2^2\right) dz \\
&= \exp\left(\frac{\|\mu_1 - \mu_2\|_2^2}{\sigma^2}\right),
\end{aligned}$$

which concludes the proof. $\square$

**Lemma C.2** (Expectation Difference Under Two Gaussians). *For Gaussian distribution $\mathcal{N}(\mu_1, \sigma^2 I)$ and $\mathcal{N}(\mu_2, \sigma^2 I)$, and for any (appropriately measurable) positive function $g$, it holds that:*

$$\mathbb{E}_{z \sim \mathcal{N}_1}[g(z)] - \mathbb{E}_{z \sim \mathcal{N}_2}[g(z)] \leq \min\left\{\frac{\|\mu_1 - \mu_2\|}{\sigma}, 1\right\} \sqrt{\mathbb{E}_{z \sim \mathcal{N}_1}[g(z)^2]}.$$

*Proof.* Define $m_i = \mathbb{E}_{z \sim \mathcal{N}_i}[g(z)]$ for $i \in \{0, 1\}$. We have:

$$\begin{aligned}
m_1 - m_2 &= \mathbb{E}_{z \sim \mathcal{N}_1}[g(z)(1 - \frac{\mathcal{N}_2(z)}{\mathcal{N}_1(z)})] \\
&\leq \sqrt{\mathbb{E}_{z \sim \mathcal{N}_1}[g(z)^2]} \sqrt{\int \frac{(\mathcal{N}_1(z) - \mathcal{N}_2(z))^2}{\mathcal{N}_1(z)} dz} \\
&= \sqrt{\mathbb{E}_{z \sim \mathcal{N}_1}[g(z)^2]} \sqrt{\exp\left(\frac{\|\mu_1 - \mu_2\|^2}{2\sigma^2}\right) - 1}
\end{aligned}$$

where we have used the previous chi-squared distance bound. Also since $m_2$ is positive,

$$m_1 - m_2 \leq m_1 \leq \sqrt{\mathbb{E}_{z \sim \mathcal{N}_1}[g(z)^2]},$$

and so

$$m_1 - m_2 \leq \sqrt{\mathbb{E}_{z \sim \mathcal{N}_1}[g(z)^2]} \sqrt{\min\left\{\exp\left(\frac{\|\mu_1 - \mu_2\|^2}{2\sigma^2}\right) - 1, 1\right\}}$$

Now if the $\min$ is not achieved by 1, then $\frac{\|\mu_1-\mu_2\|^2}{2\sigma^2} \leq 1$. And since $\exp(x) \leq 1+2x$ for $0 \leq x \leq 1$, we have:

$$\min\left\{\exp\left(\frac{\|\mu_1-\mu_2\|^2}{2\sigma^2}\right)-1,1\right\} \leq \min\left\{1+\frac{\|\mu_1-\mu_2\|^2}{\sigma^2}-1,1\right\} = \min\left\{\frac{\|\mu_1-\mu_2\|^2}{\sigma^2},1\right\}.$$

which completes the proof. $\qquad\square$

**Lemma C.3** (Self-Normalized Bound for Vector-Valued Martingales; [Abbasi-Yadkori et al., 2011]). *Let $\{\varepsilon_i\}_{i=1}^\infty$ be a real-valued stochastic process with corresponding filtration $\{\mathcal{F}_i\}_{i=1}^\infty$ such that $\varepsilon_i$ is $\mathcal{F}_i$ measurable, $\mathbb{E}[\varepsilon_i|\mathcal{F}_{i-1}] = 0$, and $\varepsilon_i$ is conditionally $\sigma$-sub-Gaussian with $\sigma \in \mathbb{R}^+$. Let $\{X_i\}_{i=1}^\infty$ be a stochastic process with $X_i \in \mathcal{H}$ (some Hilbert space) and $X_i$ being $\mathcal{F}_t$ measurable. Assume that a linear operator $V : \mathcal{H} \to \mathcal{H}$ is positive definite, i.e., $x^\top V x > 0$ for any $x \in \mathcal{H}$. For any $t$, define the linear operator $V_t = V + \sum_{i=1}^t X_i X_i^\top$ (here $xx^\top$ denotes outer-product in $\mathcal{H}$). With probability at least $1-\delta$, we have for all $t \geq 1$:*

$$\left\|\sum_{i=1}^t X_i \varepsilon_i\right\|_{V_t^{-1}}^2 \leq 2\sigma^2 \log\left(\frac{\det(V_t)^{1/2}\det(V)^{-1/2}}{\delta}\right).$$

We generalize this lemma as follows:

**Lemma C.4** (Self-Normalized Bound for Matrix-Valued Martingales). *Let $\{\varepsilon_i\}_{i=1}^\infty$ be a $d$-dimensional vector-valued stochastic process with corresponding filtration $\{\mathcal{F}_i\}_{i=1}^\infty$ such that $\varepsilon_i$ is $\mathcal{F}_i$ measurable, $\mathbb{E}[\varepsilon_i|\mathcal{F}_{i-1}] = 0$, and $\varepsilon_i$ is conditionally $\sigma$-sub-Gaussian with $\sigma \in \mathbb{R}^+$.[1] Let $\{X_i\}_{i=1}^\infty$ be a stochastic process with $X_i \in \mathcal{H}$ (some Hilbert space) and $X_i$ being $\mathcal{F}_t$ measurable. Assume that a linear operator $V : \mathcal{H} \to \mathcal{H}$ is positive definite. For any $t$, define the linear operator $V_t = V + \sum_{i=1}^t X_i X_i^\top$ Then, with probability at least $1-\delta$, we have for all $t$, we have:*

$$\left\|\sum_{i=1}^t \epsilon_i X_i^\top V_t^{-1/2}\right\|_2^2 \leq 8\sigma^2 d\log(5) + 8\sigma^2 \log\left(\frac{\det(V_t)^{1/2}\det(V)^{-1/2}}{\delta}\right)$$

*Proof.* Denote $S = \sum_{i=1}^t \epsilon_i X_i^\top$. Let us form an $\epsilon$-net, in $\ell_2$ distance, $\mathcal{C}$ over the unit ball $\{w : \|w\|_2 \leq 1, w \in \mathbb{R}^d\}$. Via a standard covering argument (e.g. [Shalev-Shwartz and Ben-David, 2014]), we can choose $\mathcal{C}$ such that $\log(|\mathcal{C}|) \leq d\log(1+2/\epsilon)$.

Consider a fixed $w \in \mathcal{C}$ and $w^\top S = \sum_{i=1}^t w^\top \epsilon_i X_i^T$. Note that $w^\top \epsilon_i$ is a $\sigma$-sub Gaussian due to $\|w\|_2 \leq 1$. Hence, Lemma C.3 implies that with probability at least $1-\delta$, for all $t$,

$$\left\|V_t^{-1/2}\sum_{i=1}^t X_i\left(w^\top \epsilon_i\right)\right\|_2 \leq \sqrt{2}\sigma\sqrt{\log\left(\frac{\det(V_t)^{1/2}\det(V)^{-1/2}}{\delta}\right)}.$$

Now apply a union bound over $\mathcal{C}$, we get that with probability at least $1-\delta$:

$$\forall w \in \mathcal{C} : \left\|V_t^{-1/2}\sum_{i=1}^t X_i\left(w^\top \epsilon_i\right)\right\|_2 \leq \sqrt{2}\sigma\sqrt{d\log(1+2/\epsilon)+\log\left(\frac{\det(V_t)^{1/2}\det(V)^{-1/2}}{\delta}\right)}.$$

For any $w$ with $\|w\|_2 \leq 1$, there exists a $w' \in \mathcal{C}$ such that $\|w-w'\|_2 \leq \epsilon$. Hence, for all $w$ such that $\|w\|_2 \leq 1$,

$$\left\|V_t^{-1/2}\sum_{i=1}^t X_i\left(w^\top \epsilon_i\right)\right\|_2 \leq \sqrt{2}\sigma\sqrt{d\log(1+2/\epsilon)+\log\left(\frac{\det(V_t)^{1/2}\det(V)^{-1/2}}{\delta}\right)}$$

$$+\epsilon\left\|\sum_{i=1}^t \epsilon_i X_i^\top V_t^{-1/2}\right\|_2.$$

By the definition of the spectral norm, this implies that:

$$\left\| \sum_{i=1}^{t} \epsilon_i X_i^\top V_t^{-1/2} \right\|_2 \leq \frac{1}{1-\epsilon} \sqrt{2} \sigma \sqrt{d \log\left(1 + 2/\epsilon\right) + \log\left( \frac{\det(V_t)^{1/2} \det(V)^{-1/2}}{\delta} \right)}$$

Taking $\epsilon = 1/2$ concludes the proof. $\qquad\square$

**Lemma C.5.** *For any sequence $x_0, \ldots x_{T-1}$ such that, for $t < T$, $x_t \in \mathbb{R}^d$ and $\|x_t\|_2 \leq B \in \mathbb{R}^+$, we have:*

$$\log \det \left( I + \frac{1}{\lambda} \sum_{t=0}^{T-1} x_t x_t^\top \right) \leq d \log \left( 1 + \frac{TB^2}{d\lambda} \right).$$

*Proof.* Denote the eigenvalues of $\sum_{t=0}^{T-1} x_t x_t^\top$ as $\sigma_1, \ldots \sigma_d$, and note:

$$\sum_{i=1}^{d} \sigma_i = \text{tr}\left( \sum_{t=0}^{T-1} x_t x_t^\top \right) \leq TB^2.$$

Using the AM-GM inequality,

$$\log \det \left( I + \frac{1}{\lambda} \sum_{t=0}^{T-1} x_t x_t^\top \right) = \log \left( \prod_{i=1}^{d} (1 + \sigma_i/\lambda) \right)$$

$$= d \log \left( \prod_{i=1}^{d} (1 + \sigma_i/\lambda) \right)^{1/d} \leq d \log \left( \frac{1}{d} \sum_{i=1}^{d} (1 + \sigma_i/\lambda) \right) \leq d \log \left( 1 + \frac{TB^2}{d\lambda} \right),$$

which concludes the proof. $\qquad\square$

## D   Simulation Setups and Results

Below, we provide simulation setups, including the details of environments and parameter settings. Specifically, the hyper-parameters, namely, 1) variance of random control variation for MPPI, 2) temperature parameter for MPPI, 3) planning horizon, 4) number of planning samples, 5) prior parameter $\lambda$, 6) posterior reshaping constant, 7) number of episodes between model updates, 8) number of features, 9) RFF bandwidth, are presented.

Note parameters were tuned in the following way: we first tuned MPPI parameters on ground truth models, then we tuned number of RFFs, their bandwidth, prior parameter, and posterior reshaping constant.

### D.1   Gym Environments

Fig. 3 plots the learning curves against GT-MPPI and the best model-based RL (MBRL) algorithm reported in Wang et al. [2019]. It is observed that $LC^3$ with RFFs quickly increased reward in early stages, indicating low sample complexities empirically.

The hyper-parameters used for InvertedPendulum, Acrobot, CartPole, Mountain Car, Reacher, and Hopper are shown in Table 2, 3, 4, 5, 6, and 7, respectively. We used JULIA_NUM_THREADS=12 for all the Gym experiments.

We mention that we tested many heuristics to improve performance such as input normalization, different prior parameter for each output dimension, using multiple bandwidth of RFFs, ensemble of RFF models, warm start of planner, experience replay, etc., however, none of them *consistently* improved the performance across tasks. Therefore we present the results with no such heuristics in this paper. Interestingly, increasing number of RFFs for some contact-rich dynamics such as Hopper did not reduce the modeling error significantly. Being able to model some of the critical interactions such as contacts seems to be the key for the success of such a complicated environment.

Table 2: Hyper-parameters used for InvertedPendulum environment.

| MPPI Hyper-parameters | Value | LC$^3$ Hyper-parameters | Value |
|---|---|---|---|
| variance of controls | $0.2^2$ | number of features | 200 |
| temperature parameter | 0.1 | RFF bandwidth | 5.5 |
| planning horizon | 10 | prior parameter | $10^{-4}$ |
| number of planning samples | 256 | posterior reshaping constant | 0 |
| | | episodes between model updates | 1 |

Table 3: Hyper-parameters used for Acrobot environment.

| MPPI Hyper-parameters | Value | LC$^3$ Hyper-parameters | Value |
|---|---|---|---|
| variance of controls | $0.2^2$ | number of features | 200 |
| temperature parameter | 0.3 | RFF bandwidth | 4.5 |
| planning horizon | 30 | prior parameter | 0.01 |
| number of planning samples | 256 | posterior reshaping constant | $10^{-3}$ |
| | | episodes between model updates | 1 |

Table 4: Hyper-parameters used for CartPole environment.

| MPPI Hyper-parameters | Value | LC$^3$ Hyper-parameters | Value |
|---|---|---|---|
| variance of controls | $0.2^2$ | number of features | 200 |
| temperature parameter | 0.1 | RFF bandwidth | 1.5 |
| planning horizon | 50 | prior parameter | $5 \times 10^{-4}$ |
| number of planning samples | 128 | posterior reshaping constant | $10^{-4}$ |
| | | episodes between model updates | 1 |

Table 5: Hyper-parameters used for Mountain Car environment.

| MPPI Hyper-parameters | Value | LC$^3$ Hyper-parameters | Value |
|---|---|---|---|
| variance of controls | $0.3^2$ | number of features | 100 |
| temperature parameter | 0.2 | RFF bandwidth | 1.3 |
| planning horizon | 110 | prior parameter | 0.01 |
| number of planning samples | 512 | posterior reshaping constant | $10^{-6}$ |
| | | episodes between model updates | 1 |

Table 6: Hyper-parameters used for Reacher environment.

| MPPI Hyper-parameters | Value | LC$^3$ Hyper-parameters | Value |
|---|---|---|---|
| variance of controls | $0.2^2$ | number of features | 300 |
| temperature parameter | 0.3 | RFF bandwidth | 4.0 |
| planning horizon | 20 | prior parameter | 0.01 |
| number of planning samples | 256 | posterior reshaping constant | 0 |
| | | episodes between model updates | 4 |

Table 7: Hyper-parameters used for Hopper environment.

| MPPI Hyper-parameters | Value | LC$^3$ Hyper-parameters | Value |
|---|---|---|---|
| variance of controls | $0.2^2$ | number of features | 200 |
| temperature parameter | 0.2 | RFF bandwidth | 12.0 |
| planning horizon | 128 | prior parameter | 0.005 |
| number of planning samples | 64 | posterior reshaping constant | 0.01 |
| | | episodes between model updates | 1 |

Figure 3: Performance curves of LC³ with RFFs for different Gym environments. Note the reward (negative cost) ranges of those plots are made different. The final mean performances of GT-MPPI and the best model-based RL (MBRL) algorithm reported in Wang et al. [2019] are also shown for reference. The algorithm is run for 200,000 timesteps and with four random seeds. The curves are averaged over the four random seeds and a window size of 5,000 timesteps.

Table 8: Hyper-parameters used for Maze environment.

| Planner Hyper-parameters | Value | $LC^3$ Hyper-parameters | Value |
|---|---|---|---|
| variance of controls | $0.3^2$ | number of features | 100 |
| temperature parameter | 0.05 | prior parameter | 0.01 |
| MPPI planning horizon | 50 | posterior reshaping constant | $10^{-3}$ (*best*) |
| MPPI planning samples | 1024 | episodes between model updates | 1 |
| PETS-CEM horizon | 50 | | |
| PETS-CEM samples | 500 | | |
| PETS-CEM elite size | 50 | | |

### D.2 Maze

In the Maze environment, states and controls are continuous and the agent plans over continuous spaces; however, the dynamics is given by 1) $x_{h+1} = x_h + [-0.5, 0]^\top$ (i.e., moving one step left) if $\lceil 2u_h \rceil = -1$, 2) $x_{h+1} = x_h + [0, -0.5]^\top$ (i.e., moving one step up) if $\lceil 2u_h \rceil = 0$, 3) $x_{h+1} = x_h + [0.5, 0]^\top$ (i.e., moving one step right) if $\lceil 2u_h \rceil = 1$, and 4) $x_{h+1} = x_h + [0, 0.5]^\top$ (i.e., moving one step down) if $\lceil 2u_h \rceil = 2$, except for the case there is a wall in the direction of travel, which then ends up $x_{h+1} = x_h$.

The hyper-parameters of Maze experiments are shown in Table 8. Note the number of features is 100 because one hot vector (e.g., $\phi(x, u) = [1, 0, \ldots, 0]^\top$ if $x \le -0.75$ and $u \le -0.5$) in this maze environment is 100 dimension. Table 8 also includes the parameters used for PETS-CEM; we used the recommended values as in the paper and the codebase, except for the planning horizon. The planning horizon was set to be the same as the MPPI counterpart. We used JULIA_NUM_THREADS=12 for all the Maze experiments.

Figure 4: Here we render a representative heatmap of the learned $W$ model from the 6 ensemble model features. Visible are 6 diagonal traces acting as a weighted average of the output of each member of the ensemble, but also significant off-diagonal values. The upper block values represent generalized positions, while the lower block is generalized velocities. Critical to modeling contact forces is accurate prediction of velocity.

## D.3 Armhand with Model Ensemble Features

This task involves a 33 DOF anthropomorphic hand at the end of a robot arm attempting to grasp and lift an object to a desired target position. We take the perspective that most model parameters of a robotic system will be known, such as kinematic lengths, actuator specifications, and inertial configurations. Since we would like robots to operate in the wild, some dynamical properties may be unknown: in this case, the manipulated object's dynamical properties. Said another way, the robot knows about itself, but only a little about the object.

In table 9, we list the dynamical properties that were randomized to make our ensemble. We use uniform distributions to present a window of possible, realistic values for the parameters: for example, we randomize the objects mass between 0.1 and 1.0 kg. The center of mass distributions is the deviation from the center of the sphere, while the moments of inertia parameter is one value applied to all elements of a diagonal inertia matrix for the object. The contact parameters are specific to the MuJoCo dynamics simulator we use Todorov et al. [2012], and are the parameters of internal contact model of the simulation. The range of values of the parameters allow for objects in the ensemble to have different softness and rebound effects. Also, table 10 lists learned model predictive error for different features, indicating that the ensemble of MuJoCo model successfully captured the true dynamics.

We clarify the setting in which this approach may be relevant as follows. Complex dynamics, such as that in the real world, are difficult to represent with function approximation like neural networks or random features. This problem can be broken into two parts: learning the structure of the features, and how to combine the features into a model. Rather than collect inordinate amounts of data to mimic the combination of features and model, we can instead use structured representations of the real world, such as dynamics simulators, to produce features, and the method in this work to learn the model. Since dynamics simulators represent the current distillation of physics into a computational form and accurate measurement of engineered systems is paramount for the modern world, this instantiation of this method is reasonable.

Table 9: Hyper-parameters used for Armhand environment.

| Hyper-parameter | Value | Ensemble Parameter | Value |
|---|---|---|---|
| variance of controls | $0.2^2$ | Models in Ensemble | 6 |
| temperature parameter | 0.08 | Mass | $\mathcal{U}(0.01, 1.0)$ |
| planning horizon | 50 | Center of Mass | $\mathcal{U}(-0.04, 0.04) \times 3$ |
| number of planning samples | 64 | Moments of Inertia | $\mathcal{U}(0.0001, 0.0004)$ |
| prior parameter | 0.0001 | Contact Param. (solimp) | $[\mathcal{U}(0.5, 0.99),$ |
| | | | $\mathcal{U}(0.4, 0.98),$ |
| | | | $\mathcal{U}(0.0001, 0.01),$ |
| | | | $\mathcal{U}(0.49, 0.51),$ |
| | | | $\mathcal{U}(1.9, 2.1)]$ |
| posterior reshaping constant | 0.01 | Contact Param. (solref) | $[\mathcal{U}(0.01, 0.03),$ |
| | | | $\mathcal{U}(0.9, 1.1)]$ |
| episodes between model updates | 1 | | |

Table 10: Learned model predictive error for different features.

| Feature method | Predictive Error: $\|x_{h+1} - W\phi\|_2/\|W\phi\|_2$ |
|---|---|
| Random Fourier Features, 2048 | 0.22 |
| 2 Layer Neural Network, 2048 hidden, $relu$ activation | 0.41 |
| Model Ensemble of 6 models | 0.09 |

## Footnotes

[1]We say a vector-valued, random variable $z$ is $\sigma$-sub-Gaussian if $w \cdot z$ is $\sigma$-sub-Gaussian for every unit vector $w$.