[Reviews · NeurIPS 2020]

Review 1

Summary and Contributions: Based upon my reading, the paper provides a mechanism for identifying a discrete-time nonlinear dynamical system by optimizing a known fitting function to obtain a kernelized approximator that induces functions in a known RKHS. The contribution of this paper is in the derivation of theoretical bounds on the cumulative regret of the learner.

Strengths: This is a very well-written paper and I enjoyed reading it. I appreciate the inclusion of code - wonderful. The bounds obtained in this work are definitely of interest to the NeurIPS community at large, and the learning-based control community in particular. I could not find any major error in the proof techniques.

Weaknesses: While the authors claim that this work connects two communities: RL and Control, I see very little evidence of control-theoretic ideas in the paper. For example, the proofs are largely agnostic to the underlying dynamics, and the focus is on ensuring bounds for an optimization problem for model-fitting rather than a proof that takes into account important behaviours such as stability and robustness (which, as a control practitioner + theorist, I posit is far more important than provably 'correct' learning). Taking nothing away from the bounds themselves - this is fantastic work in the field of statistical learning + optimization - I just don't see a clear and strong connection to the dynamics and control aspect here. Specific concerns are presented under additional comments.

Correctness: I found no critical errors in the paper other than some issues I point out under additional comments.

Clarity: Extremely.

Relation to Prior Work: The authors have certainly done a good job comparing this work with relevant research in the field that they are aware of in the RL literature. This being said, there is quite a lot of research they are unaware of in the control literature that they should consider since this is a more general paper, not just RL-centric. Some of these control theory-cognizant methods would lead, arguably, to 'safer' control as these methodologies tend to value robustness and stability over optimality/correctness. I have tried to point out a few of these papers here: This recent paper: https://arxiv.org/abs/2005.05888 does something quite similar for identifying and designing estimators for nonlinear systems but actually considers stability (safety) during the learning procedure, which is a nice bonus to have. The same author also has a paper for learning structures of nonlinear systems (e.g. Lipschitzness) from data using kernel methods (10.1109/TNNLS.2020.2978805) for guaranteeing RL performance and stability. There is also some recent work on learning nonlinear control policies by model adaptation with GPs under stability constraints: https://arxiv.org/abs/2004.02907 DOI: 10.1109/TCST.2019.2949757, DOI: 10.1109/LRA.2020.2969153 which allow for safe learning in repetitive processes, in alignment with the current paper which assumes episodes start at x_0 each time.

Reproducibility: Yes

Additional Feedback: 1. Could you please define 'continuous control' in the first sentence of the Intro? Does this mean continuous-time continuous-space control? 2. Line 20: Please could you define this term 'provably correct' as well? In control of uncertain dynamics, we rarely care about 'correctness' and far more about 'robustness' since even an arbitrarily small amount away from the exact nonlinear system could (in general) induce very different dynamics, so unless one can get an exact model, one typically relies on robust controllers for safety during implementation. 3. Line 22-23: I would recommend softening the claim that RL is "far more mature" than in the controls literature in the settings mentioned. There is a lot of work in robust control that could be (and has been) seamlessly integrated in very unknown, very safety-critical, and very complex environments (airplane/ship navigation/biomedicine) which work great without any RL. The assumptions and theorems, are, of course, quite different. I would agree with you if this said something along the lines of regret bounds or learnability results - such theory definitely does not exist in classical robust control and RL may indeed be far ahead on that curve. 4. Eqn after Line 46: I think you might need a bit more structure on your model to ensure existence and uniqueness of solutions - perhaps the RKHS assumption takes care of this somehow? Not immediately clear. 5. I see nothing in the assumptions that prevent systems with finite escape times. Unlike linear systems, nonlinear systems can have finite escape time i.e. the state blows up in finite time. What is preventing the states from growing arbitrarily large within 'H' time steps? e.g. d/dt (x) = -x^2 escapes at time = 1. I think one of the things you might need here is a stability/passivity assumption to be safe. Maybe even an explicit state-boundedness assumption. This is a gap of the current paper w.r.t. control that if clarified would make for a nice connection with the dynamical properties of the system. 6. Line 80: How does the finiteness of the dim(RKHS) tie into the control literature do you think? e.g. classical structure assumed in nonlinear control include feedback-linearizable systems, globally Lipschtiz systems etc - can a connection be made here? 7. The title of the paper says 'online' but I'm sure how you ensure you get back to x_0 in an online manner at the end of each episode? Do you assume some sort of limit cycle/repetitive behaviour? This needs to be clarified. (I assume by 'online' you mean that the system evolves naturally and you pick up the data from the state trajectories and do your fitting - is this correct? If not, please clarify.) 8. Thanks for Algo 1 - this is great. Please add a little more info in 4 - easy algorithm steps to get the BALL^t would be helpful. 9. Line 179: Asmp 2 is quite a strong assumption from a control-theoretic perspective. In essence, you are assuming the knowledge of a space of controls that are stabilizing - this is a much stronger assumption than the classical RL assumption of knowing just one stabilizing (actually, admissible i.e. inducing finite cost) control policy as in, say, policy iteration methods in the continuous-time continuous-space ADP/RL literature. 10. Asmp 2: A clarification. Is V_max largely decorative or can one actually compute this? If so the bound on Theorem 3.2 could be computed which would be great. 11. Line 198: Depending on the eigenvalues of A, Assumption 2 would not hold for 'any' linear control policy, I think. 12. Line 274: Please briefly explain what the single-traj case is. 13. I am surprised there is no persistence of excitation assumption - how can you learn a nonlinear model if your system is sitting there doing nothing? I think you might be getting this for 'free' somehow in the way that you perturb the system but no mention is made in the theory - I'm curious how this is incorporated in the identification procedure and the proofs - maybe it's hidden behind the fact that the policies selected are always such that the system does not drop to equilibrium very quickly within H steps... 14. Your practical examples have passive dynamics - I'm curious how this would do in case the model is inherently discontinuous and/or unstable. --- POST REBUTTAL --- My opinion of the paper is unchanged. I think the theoretical results are interesting enough for a score of 7.


Review 2

Summary and Contributions: This paper studies the online control problem in non-linear systems. The non-linear system is an unknown linear combination of known RKHS kernels and the proposed algorithm aims to learn this linear combination while trying to control the system. The suggested algorithm uses optimism to handle exploration vs. exploitation trade-off and achieves theoretical sqrt(T) regret the result. Due to the intractability of the optimism step, I believe Thompson Sampling is used in the experiments. It is shown that in simpler non-linear systems proposed algorithm performs well compared to benchmarks.

Strengths: -The paper is written well and the theoretical claims are sound. The analysis steps are explained thoroughly in the main text and appendix. The presentation of technical results is clear with some simple cases. -One of the first results in adaptive "non-linear" control, which is a generalization of previous linear results in the literature. -An extensive empirical evaluation is performed which gives an idea about the performance of the proposed algorithm in a more real-life setting which is both novel and promising for future research in this field.

Weaknesses: -The theoretical contribution seems marginal. Since the underlying RKHS is assumed to be known, ultimately, the system identification boils down to estimating W which follows from [1] as mentioned in the main text. Moreover, the rest of the analysis uses the Gaussian assumption on the system evolution noise in order to use GP bandit results in literature. However, even in the simple setting of LQR considered in [1], the system evolution noise is assumed to be sub-Gaussian and it is then generalized more to different noise distributions and adversarial setting. Thus, Gaussian assumption seems too restrictive. -On the contrary of the standard single trajectory result in recent online control literature, the setting is episodic and the result does not recover the known regret result in literature for LQR. I’m not sure if this is an artifact of the analysis or the unavoidable outcome of the algorithm. -In the theoretically analyzed algorithm optimism in the face of uncertainty principle is used. However, since this optimization step is computationally intractable (especially in high-dimensional control setting), it is assumed that an oracle executes this step efficiently. I’m not sure if this is a practical assumption. -In the experiments section, please correct me if I’m wrong, Thompson Sampling is used instead of optimism due to its polynomial time execution. However, even in the simpler case of LQR, Thompson Sampling has guarantees only for the scalar systems [2].

Correctness: The theoretical claims and analysis are correct. It seems like a different version of the algorithm is used in the experiments (based on Thompson sampling for optimism). It would be great, it is clarified by the authors.

Clarity: The paper is very well written and it includes discussions for special cases which improves the understanding. The experimental section, especially the details of executed algorithm, could be explained in more detail.

Relation to Prior Work: The work is well placed to the prior work in literature and somewhat generalizes the prior results in online linear control.

Reproducibility: Yes

Additional Feedback: ===================Post-Rebuttal========================= Thanks for your responses and clarifications. I've read the rebuttal and the authors somewhat clarified my questions about the comparison in Remark 3.4, Gaussian noise assumption, and the application of TS in the experiments. Hoever, I still have the following concerns: 1) I believe since the experiments are conducted using TS instead of the algorithm proposed in the paper, they don't add significant value to the overall message of the paper. 2) I still have doubts about the applicability and the significance of the results in non-linear control. By assuming the underlying RKHS is known and the unknown part is just the linear mixing matrix W, one of the biggest challenges in adaptive control of non-linear systems is bypassed (for the confidence set of W [1] is used). Moreover, by using computationally intractable optimism for the controller design, the crucial controller design process for the underlying system is agnostically avoided. There has been significant effort in linear systems to avoid this sort of black-box view of controller design and bypassing this with optimism does not give much intuition on the hardness of the problem in non-linear systems. I believe these are among the main challenges in the control theory/reinforcement learning, specifically in non-linear systems. Even though there are several significant steps and results in the paper towards adaptive control of non-linear systems, I believe the technical contribution of the paper is not fully matured yet. Therefore, I would like to keep my score at 5. ====================================================== I think this work addresses and points at a very important research direction in online control. It provides extensive realistic scenarios for empirical evaluation which will be inspiring for future developments in the area. However, in this current shape due to weaknesses mentioned above, I think the story of the paper is not complete. I believe addressing the following suggestions and questions can help this paper to provide a very significant contribution to the literature. - As mentioned by the authors, I believe considering the fully non-linear setting where the underlying RKHS is not known would complete the story of this work. With this current result, the setting seems somewhat linear still and highly similar to [1] with generalization to different underlying basis. In fact, using RFF to describe the underlying RKHS as discussed in the experiments section seems like a promising direction to consider. -I believe to make it more realistic a single trajectory setting should be considered as mentioned by the authors. The “reset” nature of the episodic setting simplifies the analysis due to simplification in data collection and avoiding the state explosion that would happen within a single trajectory which is the main problem in stabilizing the underlying system. -It is mentioned that this episodic setting brings H^2 and sqrt(d) additional factor. However, I’m not sure how this comparison works, since the settings are different. For instance, due to lack of knowledge of the system, in LQR without a stabilizing controller it is unavoidable to experience exponential growth in state size and this appears in regret expressions [1]. I believe due to episodic nature of the algorithm this is not the case in current algorithm. Similarly, Simchowitz and Foster [2020] consider a stabilizing controller and achieve the regret expression. Maybe I’m missing something but it would be good to elaborate more how the comparison in Remark 3.4 is established. -More importantly, due to intractability of optimism the algorithm seems impractical. There are various works that resolve this issue, e.g. Cohen et al. [2019] which provides SDP relaxation to the optimism step and make it polynomial time. I believe extending the current result in this direction would complete the story too. -It is not clear to me how \gamma_T(\lambda) scales in infinite dimensional feature spaces. Since it appears in the regret expression replacing the explicit dimension dependency, I believe it would be useful to provide a discussion regarding to that. -I also think it would be good to clarify the choice of optimization step in the experiments (using Thompson sampling as far as I see). -Although it may be hard to achieve with this current analysis, it would be good to consider a more general noise setting like sub-Gaussian noise or adversarial noise. [1] Y. Abbasi-Yadkori and C. Szepesva ́ri. Regret bounds for the adaptive control of linear quadratic systems. [2] M. Abbeille and A. Lazaric. Improved Regret Bounds for Thompson Sampling in Linear Quadratic Control Problems


Review 3

Summary and Contributions: The paper presents an optimistic model-based reinforcement learning algorithm for the episodic setting with dynamics that live in a RKHS and have additive Gaussian noise. It analyzes this setting theoretically and provides a sqrt{T} regret bound.

Strengths: The paper tackles a problem that is highly relevant to the NeurIPS community. There has been increased interest in model-based reinforcement learning and a theoretical understanding of which methods help in this setting is important. The paper builds up a strong theoretical foundation and develops interesting tools to analyze the regret for episodic RL with RKHS models and additive noise. In particular the tools for L3.7 are novel and interesting.

Weaknesses: The main limitations are the experiments. Numbers from PETS-CEM and PILCO are only reproduced, which makes a 1-to-1 comparison difficult. A more enlightening experiment would be to use a GP with the same features as LC3 in PILCO. This would enable a direct comparison while avoiding potential artifacts due to the model class. The experiments are also not explained clearly. While the text refers to LC3 as in Alg. 1, it sounds like at times just the mean model is taken, and at others Thompson sampling is used? Clarifying this would greatly improve the paper.

Correctness: The theoretical analysis seems correct, but I did not check every proof in detail.

Clarity: The paper is well-written overall. However, some restructuring could improve the paper: 1) Stating Theorem 1.1 before explaining what the LC3 algorithm does not seem particularly valuable. 2) In 3.1 the estimator W_t is stated as a given. It would be valuable for the reader to briefly explain the statistical properties and how it relates to W* (I.e., when is W* in Ball_t).

Relation to Prior Work: The introduction provides references for RL in general and the LQR framework in general. However, it does not mention the UCRL-class of algorithms that have been heavily investigated in the RL community. While these algorithms have largely been evaluated in the tabular setting, LC3 is in spirit basically a UCRL algorithm. Thus, this line of work should be reported on in the related work. More importantly, it seems that [1] has investigated the same problem (nonlinear control, episodic setting, sub-Gaussian noise) and derived regret bounds. While their analysis seems different in several key points, this paper should compare its analysis to the one in [1]. Note that I have no affiliation to the authors of [1] [1] http://proceedings.mlr.press/v89/chowdhury19a

Reproducibility: Yes

Additional Feedback: Concrete questions for the rebuttal: Please clarify the relationship of this paper to [1]. They seem to analyze a very similar setting. However, their bound explicitly depends on the complexity of W* (the RKHS norm). How does the bound in this paper avoid this dependency? How was LC3 implemented in the experiments? Or did the experiments actually use Thompson sampling rather than LC3? If so, what conclusions can be drawn about LC3 form the experiments? It's not obvious to me that Assumption 2 is satisfied for LQR control with any linear controller. Can't I make V_max go to infinity by picking a high-norm matrix K that destabilizes the system? So should K be bounded in remark 3.4? [1] http://proceedings.mlr.press/v89/chowdhury19a Regarding the broader impact: My understanding was to discuss potential negative implications, but the current impact statement is only mentioning positive aspect. ----------------- After rebuttal: My opinion of the paper is largely unchanged. I think the theoretical results are sufficient for publication at NeurIPS. The main weak point remains for me in the experiment section: They are fairly simplistic and while maybe one can conclude that "exploration" helps in these settings, TS can experimentally behave quite different from UCB.

[Author Response · NeurIPS 2020]

We thank reviewers for carefully reviewing our paper and providing constructive feedback.

**Responses to Reviewer 1.** We thank the reviewer for pointing to related control papers and providing a perspective
from the control theory side. We will discuss these related works in the revised version. Stability and safety are indeed
important, and it may be possible to incorporate such constraints into our framework, where our uncertainty estimates
can characterize safety/stability constraints and enable more cautious exploration. We think this is an important future
direction. We will soften some of our claims and state that we focus on optimality (in the no-regret sense).

Due to space limit, below we response to a few main questions from R1 while addressing all questions in the revised
version: (1) We consider discrete time, continuous space control; (2) "Provably correct" means no-regret with respect to
the best policy in $\Pi$ on the real system; (3) We didn't explicitly assume boundness on states as under Gaussian noise,
states' norm are not bounded—instead, we use Asmp 2 and info-Gain $\gamma_T$: our regret bound still holds as long as $V_{\max}$
and $\gamma_T$ are bounded even though states could be large. (4) the reviewer's interpretation of "online" is correct; (5) Asmp
2 is actually weaker than the boundedness cost/reward assumption in almost all theoretical RL regret analysis; (6)
Specializing to LQR, Asmp 2 implies that $\Pi$ is a subset of stabilizing policies (e.g., see Cohen et.al 19 for LQR case).

**Responses to Reviewer 2.** KNR is fundamentally different from LQR, as it can capture nonlinear $f$ such as Piecewise
Affine System, i.e., hybrid system, and higher-order polynomials. The consequences of this generalization are:

1. Planning itself (even non-optimistic planning) is computationally intractable. This means there is really no hope for
provably computationally efficient algorithms, and heuristics must be required in practice.
2. Smoothness, which is understood to be an important aspect of continuous controls, is not a prior inherent in KNR
as it is in the LQR. Gaussian noise provides smoothness, although other explicit smoothness assumptions (e.g.,
Smoothness w.r.t value functions [Osband & Van Roy 14]) can generalize to sub-Gaussian noises.
3. LQR's value function is quadratic and smooth (under stable linear $\pi$) which is leveraged in the LQR's analysis.
KNR's value functions can be complicated and we need to derive the self-bounding lemma 3.7 (with a novel
application of optional stopping time argument) to use the second moment of the realized total cost.
4. Our analysis is far more than GP-bandit: as we do not assume the cost is bounded as most bandit/RL works did, this
requires a key new technique: the self-bounding lemma (Lemma 3.7). This new lemma also enables us to get a
first-order regret bound scaling with $J^\star$ (Thm 3.6) which was missing even in prior LQR works.
5. Full nonlinear $f(x, u)$ without further structural assumption is not tractable even in statistics, as this generalizes
infinite arm bandit problems.

*Episodic vs single-trajectory*: we believe our results can extend to single trajectory setting under stronger assumptions
such as strongly stable system with stable controllers (e.g., Cohen et.al 19), and this is a direction for future work. If the
closed-loop system is strongly stable, then single trajectory setting is similar to episodic setting, as the dependency
between the current state and the previous states are diminishing exponentially fast (e.g., see Hazan et.al 19).There are
already challenges in the finite horizon, which is relevant for many practical settings work. This is related to Remark
3.4 which we will make precise in the final version (along with the $d$ factors); we will make a more careful elaboration
based on the system parameters as how these factors scale (based on the assumptions in [Simchowitz and Foster [2020]).

Instead of directly assuming strongly stable system (Cohen et.al 19), we assume that any policy in the policy class $\Pi$
has bounded second moment of their realized total cost (Assumption 2). When specializing to LQR, such strongly
stable LQR systems imply our assumption, and hence our assumption is more general.

*Regarding $\gamma_T$*: we will provide $\gamma_T$ for popular kernels such as RBF and Matérn. for RBF, it scales $O(\log(HT)^{d_x+d_a+1})$.

*Regarding Thompson sampling*: we use TS in experiments as it's a simple alternative of UCB-based approach. We
do believe that by leveraging the framework from Russo and Van Roy [2014], we can obtain a Bayesian regret bound.
[2] presents a frequentist regret bound for 1-d LQR. Frequentist regret bound for KNR is challenging due to the value
function of KNR can be complicated. A frequentist analysis of TS for KNR is an interesting future work.

**Responses to Reviewer 3.** The PILCO implementation from Wang et.al uses GP with RBF kernel while our
implementation uses RFF feature corresponding to RBF kernel. So the model class capacity is similar.

We thank the reviewer for pointing us to a related work. The major difference is that [1] assumes Lipschitzness in the
one-step future value function (same as Osband and Van Roy 14). As we do not assume boundness on the realized total
cost, such Lipschitz constant can be unbounded making the theorem in [1] vacuous. Indeed, it is this relaxation forces
us to develop a novel technique (Lemma 3.9) so that our results do not require such Lipschitzness assumption.

We use Thompson sampling (TS) in the experiments as a simple alternative of UCB-based approach. The mean model
is used as a baseline to show exploration via TS (based on the uncertainty ball from LC$^3$) helps. We require any $\pi$ in
$\Pi$ to satisfy assumption 2. Mapping to LQR, this means that $\Pi$ could set to be a subset of all linear controllers that
contains only stabilizing linear controllers which is a common assumption used in LQR analysis (e.g., Cohen et.al 19).

[Meta-Review · NeurIPS 2020]

Paper concerns sequential control of a nonlinear dynamical system with the underlying dynamics being a function in RKHS. The introduced algorithm LC3 enjoys an O(sqrt{T}) regret bound against the optimal controller with no explicit dependence on the dimension of the system dynamics. The paper received a mostly positive evaluation from the reviewers with one vote below the acceptance threshold (scores of 7, 7, and 5). The main strengths of the paper were identified as: - Novel results (on of the first in adaptive non-linear control) which should be of interest to the NeurIPS community. - The paper is very well-written, the technical quality is sound, and the code was included in the supplement. - Extensive empirical evaluation (although one of the reviewers found the evaluation to be inappropriate). Several weaknesses were also pointed out: - One of the reviewers found the contribution of the theoretical results to be marginal comparing to the past work. - Episodic setting contrary to standard single trajectory results in recent online control literature. - Intractability of the optimization step. - Little evidence of control-theoretic ideas in the paper.